# Methane production by three widespread marine phytoplankton species: release rates, precursor compounds, and potential relevance for the environment

Thomas Klintzsch[1*], Gerald Langer[2], Gernot Nehrke[3], Anna Wieland[1], Katharina Lenhart[1,4] and Frank Keppler[1,5*]

[1]Institute of Earth Sciences, University Heidelberg, Im Neuenheimer Feld 234-236, 69120 Heidelberg, Germany.
[2]The Marine Biological Association of the United Kingdom, The Laboratory, Citadel Hill, Plymouth,
Devon, PL1 2PB, UK
[3]Marine Biogeosciences, Alfred Wegener Institut – Helmholtz-Zentrum für Polar- und Meeresforschung, Bremerhaven, Germany
[4]University of Applied Sciences, Berlinstr. 109, Bingen 55411, Germany
[5]Heidelberg Center for the Environment HCE, Heidelberg University, D-69120 Heidelberg, Germany

* Correspondence to: Thomas Klintzsch (thomas.klintzsch@geow.uni-heidelberg.de; Tel: +49 6221 546006), Frank Keppler (frank.keppler@geow.uni-heidelberg.de; Tel: +49 6221 546009)

**Abstract.** Methane ($CH_4$) production within the oceanic mixed layer is a widespread phenomenon, but the underlying mechanisms are still under debate. Marine algae might contribute to the observed $CH_4$ oversaturation in oxic waters, but so far direct evidence for $CH_4$ production by marine algae has only been provided for the coccolithophore *Emiliania huxleyi*.

In the present study we investigated, next to *Emiliania huxleyi*, other widespread haptophytes, i.e. *Phaeocystis globosa* and *Chrysochromulina sp.*. We performed $CH_4$ production and stable carbon isotope measurements and provide unambiguous evidence that all three investigated marine algae are involved in the production of $CH_4$ under oxic conditions. Rates ranged from $1.9 \pm 0.6$ to $3.1 \pm 0.4$ µg $CH_4$ per g POC (particulate organic carbon) $d^{-1}$ with *Chrysochromulina sp.* and *Emiliania huxleyi* showing the lowest and highest rates, respectively. Cellular $CH_4$ production rates ranged from $16.8 \pm 6.5$ (*Phaeocystis globosa*)

to $62.3 \pm 6.4$ ag $CH_4$ $cell^{-1}$ $d^{-1}$ (*Emiliania huxleyi*; ag = $10^{-18}$ g). In cultures that were treated with $^{13}$C-labelled hydrogen carbonate $\delta^{13}CH_4$ values increased with incubation time, resulting from the conversion of $^{13}$C-hydrogen carbonate to $^{13}CH_4$. The addition of $^{13}$C labelled dimethyl sulfide, dimethyl sulfoxide and methionine sulfoxide – known algal metabolites that are ubiquitous in marine surface layers – resulted in the occurrence of $^{13}$C-enriched $CH_4$ in cultures of *Emiliania huxleyi* clearly indicating that methylated sulphur compounds are also precursors of $CH_4$. By comparing the algal $CH_4$ production rates from our laboratory

experiments with results previously reported in two field studies of the Pacific Ocean and the Baltic Sea we might conclude that algae mediated $CH_4$ release is contributing to $CH_4$ oversaturation in oxic waters. Therefore, we propose that haptophyte mediated $CH_4$ production could be a common and important process in marine surface waters.

## 1. Introduction

Methane ($CH_4$), the second most important anthropogenic greenhouse gas after $CO_2$, is the most abundant reduced organic compound in the atmosphere and plays a central role in atmospheric chemistry (Denman et al., 2007; Kirschke et al., 2013; Lelieveld et al., 1998). The mixing ratio of $CH_4$ in the atmosphere has been increasing dramatically from pre-industrial values of about 715 parts per billion by volume (ppbv) to about 1868 ppbv (October 2018, NOAA). The global atmospheric $CH_4$ budget is determined by the total emission (540-568 Tg $CH_4$ $yr^{-1}$) of various sources from terrestrial and aquatic surface areas, that are balanced primarily by one major sink (hydroxyl radicals) in the atmosphere. The world's oceans are considered to be a minor source of $CH_4$ to the atmosphere (1-3 %, Saunois et al., 2016). However, in recent years the widespread occurrence of in situ $CH_4$ production in the ocean mixed layer has received much attention, since $CH_4$ formation in the oxygenated ocean mixed layer challenge the paradigm that biological methanogenesis is a strictly anaerobic process.

Methane is primarily formed by degradation of buried organic matter under heat and pressure (thermogenic) inside the earth crust or produced by the incomplete combustion of biomass (pyrogenic). On the other hand, $CH_4$ resulting from microbial processes, carried out by methanogenic archaea under anoxic conditions in soils and sediments or the digestion system of ruminants are categorized as biogenic or microbial (Kirschke et al., 2013). In contrast to these well-known sources, recent studies have confirmed direct $CH_4$ release from eukaryotes, including plants, animals, fungi, lichens, and the marine alga *E. huxleyi* even in the absence of methanogenic archaea and in the presence of oxygen or other oxidants (Keppler et al., 2006; Ghyczy et al., 2008; Lenhart et al., 2012; Lenhart et al., 2016; Lenhart et al., 2015b). A very recent study also confirmed *cyanobacteria*, as $CH_4$ producers suggesting that $CH_4$ production occurs in all three domains of life (Bizic-Ionescu et al., 2018). These novel sources, from the domains *eucarya* and *bacteria,* might be classified as biotic non-archaeal $CH_4$ (Boros and Keppler, 2018).

In situ $CH_4$ production in oxygenated surface waters in the marine environment was first reported by Scranton and Farrington (1977) and Scranton and Brewer (1977) and some decades later also for lakes (Grossart et al., 2011). Significant quantities of $CH_4$, produced in upper oxic waters, near the air-water interface, might overcome oxidation, and thus significantly contributing to $CH_4$ fluxes from aquatic environments to the atmosphere (Bogard et al., 2014). It turned out that in situ $CH_4$ production in the upper oxic waters is a common feature of both oceans and lakes (Forster et al., 2009; Reeburgh, 2007; Tang et al., 2014;Donis et al., 2017; Bižić-Ionescu et al., 2018; Bange et al., 1994). These results have stimulated the scientific community to study in more detail the phenomenon of $CH_4$ occurrence in oxygenated surface waters. In this context, emissions from cyanobacteria or algae might help to explain the phenomenon of dissolved $CH_4$ oversaturation. In addition, it has been suggested that $CH_4$ might be produced by the bacterial cleavage of methylphosphonate (MPn) in oligotrophic marine Pacific waters during phosphorus limitation (Karl et al., 2008; Metcalf et al., 2012; Repeta et al., 2016). While dissolved MPn in surface waters cannot account for the $CH_4$ oversaturation observed in the oligotrophic waters of the North Pacific (Valle and Karl, 2014), the cycling of the organic matter phosphonate inventory might be sufficient to support the total atmospheric $CH_4$ flux (Repeta et al., 2016).

In contrast to this apparently non-oxygen sensitive pathway, many other studies have identified the "traditionally" archaeal methanogenesis in anoxic microenvironments as a $CH_4$ source. Floating particles (Karl and Tilbrook, 1994), the digestive tracts of zooplankton (de Angelis and Lee, 1994; Stawiarski et al., 2019; Schmale et al., 2018) or fishes (Oremland, 1979) have been found as anoxic micro niches for methanogens. It has been suggested that some methanogens might be active under

oxic conditions by being equipped with enzymes to counteract the effects of molecular oxygen during methanogenesis (Angel et al., 2011). Potential substrates for methylotrophic methanogens in such micro niches are the algae metabolites dimethylsulfoniopropionate (DMSP) and their degradation products dimethyl sulfide (DMS) or dimethyl sulfoxide (DMSO) (Zindler et al., 2013; Damm et al., 2008; Florez-Leiva et al., 2013). Furthermore, DMSP might also be converted to $CH_4$ by nitrogen limited bacteria (Damm et al., 2010; Damm et al., 2015). However, in coastal waters where DMS and DMSP

production is enhanced, $CH_4$ was found to mainly related to sedimentary sources (Borges et al., 2018).

In contrast to microbial processes, which are considered to be driven by enzymes, $CH_4$ might also be derived by the chemical reaction of chromophoric dissolved organic matter (CDOM) and DMS induced by UV or visible light under both oxic and anoxic conditions (Zhang et al., 2015). A similar photochemical $CH_4$ formation was earlier described for acetone by Bange and Uher (2005) but the production of $CH_4$ from acetone was considered negligible under oxic conditions.

Another chemical reaction that readily forms $CH_4$ from the methyl thioethers and their sulphoxides under highly oxidative conditions and catalyzed by nonheme iron-oxo (IV) species was presented by Althoff et al. (2014) and Benzing et al. (2017). Iron-oxo species have been identified as active intermediates in the catalytic cycles of a number of biological enzymatic systems (Hohenberger et al., 2012). Thus, marine algae containing elevated concentrations of methyl thiolethers and their sulfoxides such as DMSP, DMSO, methionine (MET) or methionine sulphoxide (MSO), might be biochemical reactors for

non-archaeal $CH_4$ production as it was already proposed by Lenhart et al. (2016) and Keppler et al. (2009).

Marine phytoplankton plays a central role in the global carbon cycle: Approximately a half of earth's primary production is carried out by marine phytoplankton (Field et al., 1998). In this context it is important to mention that almost 40 years ago researchers (Scranton and Brewer, 1977; Scranton and Farrington, 1977; Scranton 1977) already mentioned the possibility of in-situ formation of $CH_4$ by marine algae, since $CH_4$ production was examined in cultures of *E. huxleyi* and *T. pseudonana*.

Furthermore, a direct isotopic evidence for $CH_4$ production by marine algae in the absence of methanogenic archaea has only been provided for *E. huxleyi* (Lenhart et al., 2016). Based on the application of stable carbon isotope techniques, it could be clearly shown that both hydrogen carbonate and a position-specific [13]C-labelled MET were carbon precursors of the observed $CH_4$ production. However, it remains unclear whether $CH_4$ production also occurs among other marine algae and if there are also other carbon precursors, involved in the formation process.

In the present study we investigated, next to the coccolithophore *E. huxleyi*, two other marine, non-calcifying *haptophy*tes, namely *P. globosa* and *Chrysochromulina sp.* for $CH_4$ formation. The investigated species are all bloom-forming and often found as dominant members in marine phytoplankton community worldwide (Schoemann et al., 2005; Thomsen, 1994; Brown and Yoder, 1994). Furthermore, they are well-known for their high DMSP, DMS and DMSO productivity (Liss et al., 1994; Keller, 1989; Holligan et al., 1993; Stefels et al., 2007; Matrai and Keller, 1993). We therefore conducted stable isotope

experiments using $^{13}C$ labelled DMS, MSO and DMSO to identify potential methyl group precursor compounds that eventually lead to $CH_4$ production. Finally, we discuss the laboratory $CH_4$ production rates in relation to its potential significance in marine environments.

## 2.    Material & Methods

### 2.1  Cultures and culture conditions

Three algal species, *Emiliania huxleyi* RCC1216 obtained from the Roscoff Culture Collection (http://roscoff-culture-collection.org/) *Phaeocystis globosa* PLY 575 and *Chrysochromulina sp.* PLY 307 obtained from the Marine Biological Association of the United Kingdom (https://www.mba.ac.uk/facilities/culture-collection) were studied. In order to keep non-axenic algae cultures largely free of bacteria, the cultures were diluted regularly, resulting in quasi constant exponential algal
growth while minimizing bacterial cell density.

All incubation experiments were carried out in controlled and sterile laboratory conditions under a 16/8 hour light/dark cycle at a light intensity of 350 µmol photons $m^{-2}$ $s^{-1}$ and a temperature of 20°C. All samples were taken at the end of the light cycle. Monoclonal cultures were grown in full-batch mode (Langer et al., 2013) in sterile filtered (0.2 µm Ø pore size) natural North Sea seawater (sampled off Helgoland, Germany) enriched in nutrients according to F/2 medium (Guillard and Ryther, 1962).
The initial dissolved inorganic carbon (DIC) of the F/2 medium was $2152 \pm 6$ µmol $L^{-1}$ (measured by Shimadzu TOC-V CPH). The DIC value falls within the range of typical DIC concentrations of North Sea seawater.

### 2.2  Determination of cell densities

Cell densities were determined from four aliquots of each culture sample, using either a Fuschs-Rosenthal or Neubauer counting chamber, depending on cell density.

### 2.3  Incubation with $^{13}C$- labelled hydrogen carbonate

To investigate $CH_4$ production by algal cultures borosilicate glass bottles (Schott, Germany) filled with 2.0 L 0.2 µm filtered F/2 medium and with 0.35 L headspace volume were used in our investigations of C*hrysochromulina sp.* and *P. globosa*. For the investigations of *E. huxleyi* 0.85 L medium and 0.4 L headspace volume were used (Schott, Germany). The vails were sealed airtight with lids (GL 45, PP, 2 port, Duran Group) equipped with one three-way port for liquid and a second port fitted
with a septum for gas sampling. For measurements of the mixing ratio and stable carbon isotope value of $CH_4$ ($\delta^{13}C$-$CH_4$)

samples of headspace (20 mL) were taken from each vial. Afterwards, samples (2 mL) for determining cell densities were taken. In order to maintain atmospheric pressure within the vial, the surrounding air was allowed to enter via the three-way port and trough a sterile filter to avoid biological contamination. The inflow of surrounding air was taken into consideration when $CH_4$ production was calculated.

Cultures that were studied during the incubation were inoculated from a pre-culture grown in dilute-batch mode (Langer et al., 2009). To investigate algal derived $CH_4$ formation six vials were inoculated with algae and another six vials contained medium only.

In addition, three vials of each group were treated with $^{13}C$-hydrogen carbonate ($H^{13}CO_3^-$) to investigate $CH_4$ formation by measuring stable carbon isotope values of $CH_4$. Four different treatments were used: medium either with $H^{13}CO_3^-$ (medium +

$H^{13}CO_3^-$) or without (medium, data not available) and cultures supplemented either with $H^{13}CO_3^-$ (medium + culture + $H^{13}CO_3^-$) or without (medium + culture). The different treatments and the number of replicates for the experiments with C*hrysochromulina sp.* and *P. globosa* are provided in Fig. 1.

Please note that stable isotope measurements using $H^{13}CO_3^-$ were not performed for *E. huxleyi* as evidence for isotope labelling of $CH_4$ was already provided by Lenhart et al. (2016). To study $CH_4$ formation of *E. huxleyi* by measuring headspace

concentration three replicates (culture and medium group, n=3) were used.

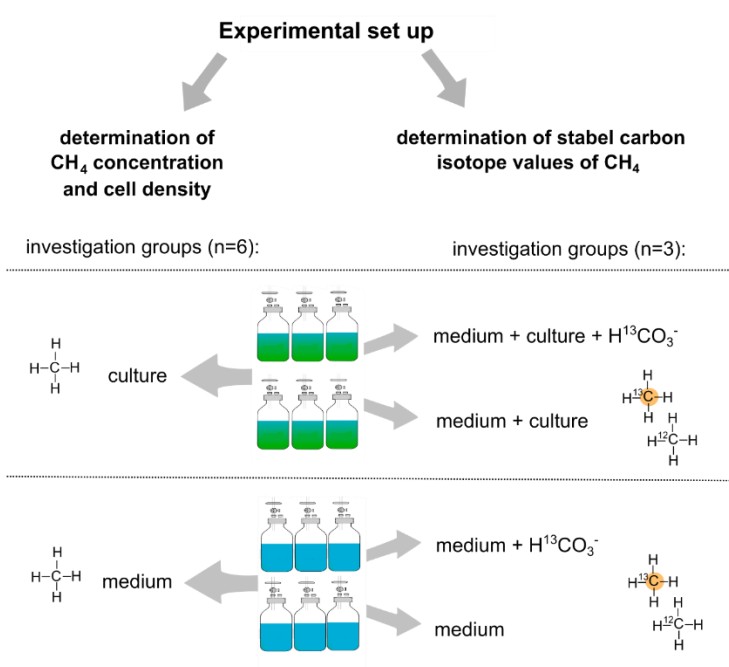

**Fig. 1 Experimental setup for measuring $CH_4$ formation by *Chrysochromulina sp.* and *P. globosa*. Methane formation was investigated by concentration measurements within six vials containing either algae or medium only (left column). For stable isotope measurements of $CH_4$ $^{13}C$ labelled hydrogen carbonate ($H^{13}CO_3^-$) was added to three vials of both groups (right column).**


The overall incubation time was 9, 11 and 6 days for *Chrysochromulina sp., P. globosa* and *E. huxleyi* respectively. Headspace and liquid samples were collected on a daily basis for *E. huxleyi* and in 2-3 days intervals from cultures of *Chrysochromulina sp.* and *P. globosa*. The incubation time and sampling intervals varied between species because of variations in the growth rate and the cell density in the stationary phase. Cell densities were plotted versus time and the exponential growth rate ($\mu$) was calculated from exponential regression using the natural logarithm (Langer et al., 2013). The exponential growth phase (from which $\mu$ was calculated) was defined by the cell densities which corresponded to the best fit ($r^2 > 0.99$) of the exponential regression. This was done by using the first three (*Chrysochromulina sp.* and *E. huxleyi*) or four data points (*P. globosa*) of the growth curve.

For stable carbon isotope experiments 48,7 µmol $L^{-1}$ $NaH^{13}CO_3$ in final concentration was added to the F/2 medium. The added amount of $NaH^{13}CO_3$ corresponds to 2% of the DIC of the North Sea seawater ($2152 \pm 6$ µmol $L^{-1}$), resulting in a theoretically calculated $\delta^{13}C$ value of DIC of $+2014 \pm 331$‰. To determine the $\delta^{13}C$-$CH_4$ values of the source, the Keeling-plot method was applied (Keeling, 1958). For a detailed discussion of the Keeling plot method for determination of the isotope ratio of $CH_4$ in environmental applications, please refer to (Keppler et al., 2016). Oxygen concentration was monitored daily (using inline oxygen sensor probes, PreSens, Regensburg) at the end of the light cycle (Fig. S1).

## 2.4 Determination of $CH_4$ production rates

Since the experiment in the section 2.3 was not designed to obtain POC quotas (POC = particulate organic carbon), we conducted an additional experiment. To best compare $CH_4$ formation rates of the three algae species it is necessary to obtain exponential growth to ensure constant growth rates and constant (at a given time of day) cellular POC quotas over the course of the experiment. Exponential growth is a prerequisite for calculating production on the basis of growth rate and quota (here $CH_4$ quota). The point is a general, technical one, and is not confined to $CH_4$ production. The studies by Langer et al. (2012, 2013) discuss this point in the context of batch culture experiments. Briefly, production on this account is the product of growth rate and quota (e.g. $CH_4$, calcite, organic carbon). Production here is an integrated value, typically over many cell divisions. For this calculation of production to be meaningful a constant growth rate is required. The exponential growth phase fulfills this criterion whereas the transition phase and the stationary phase do not. Therefore production cannot be calculated meaningfully in the non-exponential phases. The problem can, however, be minimized by using small increments (one day) because growth rate can be regarded as quasi-constant (see also Lenhart et al., 2016). The $CH_4$ production rates can be calculated by multiplying the growth rate $\mu$ with the corresponding cellular or POC-$CH_4$ quota, that was measured at the end of the experiment.

For this additional experiment the cultures were grown in 160 mL crimped serum bottles filled with 140 mL medium and 20 mL headspace (n=4). Oxygen concentration was monitored (using inline oxygen sensor probes, PreSens, Regensburg) at the end of each light and dark cycle (Fig. S2).

The growth rate ($\mu$) was calculated from cell densities of the beginning and end of the experiment according to Eq.1:

$$\mu = \frac{Ln(N_1) - Ln(N_0)}{(t_1 - t_0)} \tag{1}$$

where $N_0$ and $N_1$ are the cell densities at the beginning ($t_0$) and end of the experiment ($t_1$). The daily cellular $CH_4$ production rates ($CH_4P_{cell}$, ag $CH_4$ cell$^{-1}$ d$^{-1}$, ag = $10^{-18}$ g) were calculated according to Eq.2:

$$CH_4P_{cell} = \mu \times \frac{m(CH_4)}{cell} \tag{2}$$

where $m(CH_4)$ is the amount of $CH_4$ that was produced at the end of the experiment.

To calculate POC based $CH_4$ production rates the cellular organic carbon content ($POC_{cell}$) was derived from cell volume ($V_{cell}$) by using the Eq. 3 according to Menden-Deuer and Lessard (2000):

$$POC_{cell} = 0.216 \times V_{Cell}^{0.939} \tag{3}$$

The cell volume was determined measuring the cell diameter in light micrographs using the program ImageJ (Schindelin et al., 2012).

According to (Olenina, 2006) a ball shape can be assumed for calculating the cell volume for the three species investigated here. The daily cellular $CH_4$ production rates ($CH_4P_{POC}$, µg $CH_4$ g$^{-1}$ POC d$^{-1}$) were calculated from growth rate and $CH_4$-POC quotas at the end of the experiment according to Eq. 4.

$$CH_4P_{POC} = \mu \times \frac{m(CH_4)}{POC} \tag{4}$$

The $CH_4$ production potential ($CH_4$-PP) was used to translate differences in cellular production rates to community level. According to Gafar et al. (2018), the $CH_4$-PP can be calculated for different periods of growth, by calculating a cellular standing stock for each time period from a known starting cell density ($N_0$) (whereby constant exponential growth is assumed). The corresponding amount of produced $CH_4$ ($CH_4$PP) for each period of growth and standing stock is the product of the cellular standing stock and $CH_4$ quota (Eq. 5).

$$CH_4PP = N_0 \times e^{\mu \times t} \times \frac{m(CH_4)}{cell} \tag{5}$$

In the present study the $CH_4$-PP was calculated for a standing stock that is obtained after 7 days of growth starting with a single cell.

## 2.5 Incubation with $^{13}$C labelled DMS, DMSO and MSO

The sulphur bonded methyl group(s) in DMS, DMSO and MSO were investigated as precursors for algal-derived $CH_4$ in an incubation experiment with *E. huxleyi*. For all tested compounds only the C atom of the sulphur bonded methyl group(s) was labelled with $^{13}$C (R-S-$^{13}$CH$_3$, 99%). A final concentration of 10 µM were used for each compound.

The different treatments to investigate potential $CH_4$ formation by $^{13}C_2$-DMS, $^{13}C_2$-DMSO, $^{13}C$-MSO are provided in Fig. 2.Three independent replicates and repeated measurements over time were used. Headspace and vail size were analogous to the experiment described in section 2.3 for *E. huxleyi*. Samples were taken daily during an overall incubation time of 6 days.

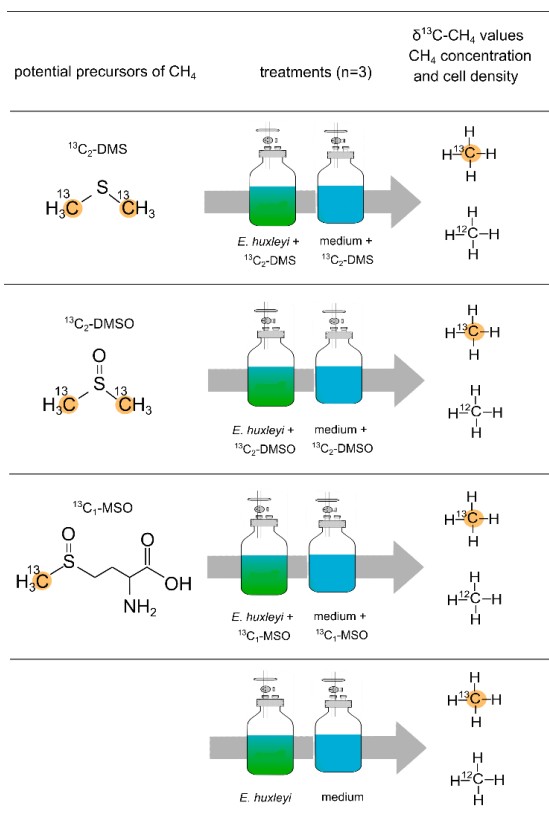

**Fig. 2 Experimental setup to investigate potential precursor compounds of $CH_4$. Dimethyl sulfide ($^{13}C_2$-DMS), dimethyl sulfoxide ($^{13}C_2$-DMSO) and methionine sulfoxide ($^{13}C$-MSO) were added to the vials containing either a culture of *E. huxleyi* or medium only. For all tested compounds only the carbon atom of the sulphur bonded methyl group(s) was labelled with $^{13}C$.**

## 2.6 Determination of $CH_4$ mass

Five mL of a gas sample was collected from the head space of the vials using a gas tight Hamilton gas syringe. The sample was analyzed by gas chromatography (GC-14B, Shimadzu, Japan; column: 2 m, Ø = 3.175 mm inner diameter, high-grade steel tube packed with Molecular Sieve 5A 60/80 mesh from Supelco) equipped with a flame ionization detector (FID). Quantification of $CH_4$ was carried out by comparison of the integrals of the peaks eluting at the same retention time as that of the $CH_4$ authentic standard, using two reference standards containing 9837 and 2192 parts per billion by volume (p.p.b.v).

Mixing ratios were corrected for head space pressure that was monitored using a pressure measuring device (GMSD 1,3 BA, Greisinger).


The CH₄ mass $\left(m_{CH_4}\right)$ was determined by its mixing ratio $\left(x_{CH_4}\right)$ and the ideal gas law (Eq. 6),

$$m_{CH_4} = M_{CH_4} \times x_{CH_4} \frac{p \times V}{R \times T}$$

(6)

where $M_{CH_4}$ = molar mass, p = pressure, T = temperature, R = ideal gas constant, V = volume.

The dissolved CH₄ concentration was calculated by using the equation of Wiesenburg and Guinasso (1979).

## 2.7 GC-C-IRMS measurements


Stable carbon isotope values of CH₄ of headspace samples were analyzed by gas chromatography stable isotope ratio mass spectrometry (GC-C-IRMS, Deltaplus XL, Thermo Finnigan, Bremen, Germany). All $\delta^{13}$C-CH₄ values were corrected using two CH₄ working standards (isometric instruments, Victoria, Canada) with values of $-23.9 \pm 0.2$‰ and $-54.5 \pm 0.2$‰. The results were normalized by two-scale anchor calibration according to (Paul et al., 2007). The average standard deviation of the analytical measurements was in the range of 0.1 ‰ to 0.3 ‰ (based on three repeated measurements of CH₄ working


standards). All $\delta^{13}$C-CH₄ values are expressed in the conventional $\delta$ notation, in per mille (‰) vs. Vienna Pee Dee Belemnite (VPDB), using Eq.7.

$$\delta^{13}C = \frac{\left(\frac{^{13}C}{^{12}C}\right)_{sample}}{\left(\frac{^{13}C}{^{12}C}\right)_{standard}} - 1$$

(7)

For a detailed description of the $\delta^{13}$C-CH₄ measurements by GC-IRMS and technical details of the pre-concentration system we would like to refer to previous studies by (Comba et al., 2018) and (Laukenmann et al., 2010)

## 2.8 Statistics


To test for significant differences in cell density, CH₄ formation, and CH₄ content between the treatments, two-way analysis of variance (ANOVA) (considering repeated measurements) and a post hoc test [Fisher least significant difference (LSD) test; alpha 5 %] were used.

## 3. Results

### 3.1 Algal growth and CH$_4$ formation

To investigate CH$_4$ production by algal cultures incubations with $^{13}$C-labelled hydrogen carbon were applied as described in section 2.3. The growth curves during incubation of the three algal species are presented in Fig. 3 (upper panel a, b, c). The initial cell densities were $26.9 \pm 4.0 \times 10^3$ cells mL$^{-1}$ for C$hrysochromulina$ $sp.$, $25.6 \pm 1.2 \times 10^3$ cells mL$^{-1}$ for $P.$ $globosa$ and $17.5 \pm 2.0 \times 10^3$ cells mL$^{-1}$ for $E.$ $huxleyi$. The exponential growth rate $\mu$ was highest for $E.$ $huxleyi$ ($1.71 \pm 0.04$ d$^{-1}$) i.e. three or five times higher than for $P.$ $globosa$ and C$hrysochromulina$ $sp.$ (with $0.33 \pm 0.08$ d$^{-1}$ and $0.52 \pm 0.07$ d$^{-1}$, respectively). The

dotted lines in Fig. 1 a, b, c marks the time points of exponential growth.

Maximum cell densities were lowest for C$hrysochromulina$ $sp.$ with $0.18 \pm 0.01 \times 10^6$ cells mL$^{-1}$ followed by $E.$ $huxleyi$ with $1.70 \pm 0.09 \times 10^6$ cells mL$^{-1}$ and highest for $P.$ $globosa$ with $1.77 \pm 0.15 \times 10^6$ cells mL$^{-1}$.

Significant CH$_4$ formation was observed in all three cultures over the whole incubation period of 5 to 11 days (Fig. 3 d, e, f) whereas no increase in CH$_4$ over time was observed in the control groups. For all species the increase in headspace CH$_4$ was

significant ($p \leq 0.05$) at second time point of measurement and at all following time points ($p \leq 0.001$). At the end of the incubation period the amounts of produced CH$_4$ were $34.9 \pm 7.3$ ng, $99.3 \pm 8.2$ ng and $45.0 \pm 3.1$ ng for $Chrysohromulina.$ $sp.$, $P.$ $globosa$ and $E.$ $huxleyi$, respectively. A linear correlation was found between the absolute number of cells and the amount of produced CH$_4$ of $Chrysochromulina$ $sp.$, $P.$ $globosa$ and $E.$ $huxleyi$ (Fig. 3 lower panel, g, h, i).

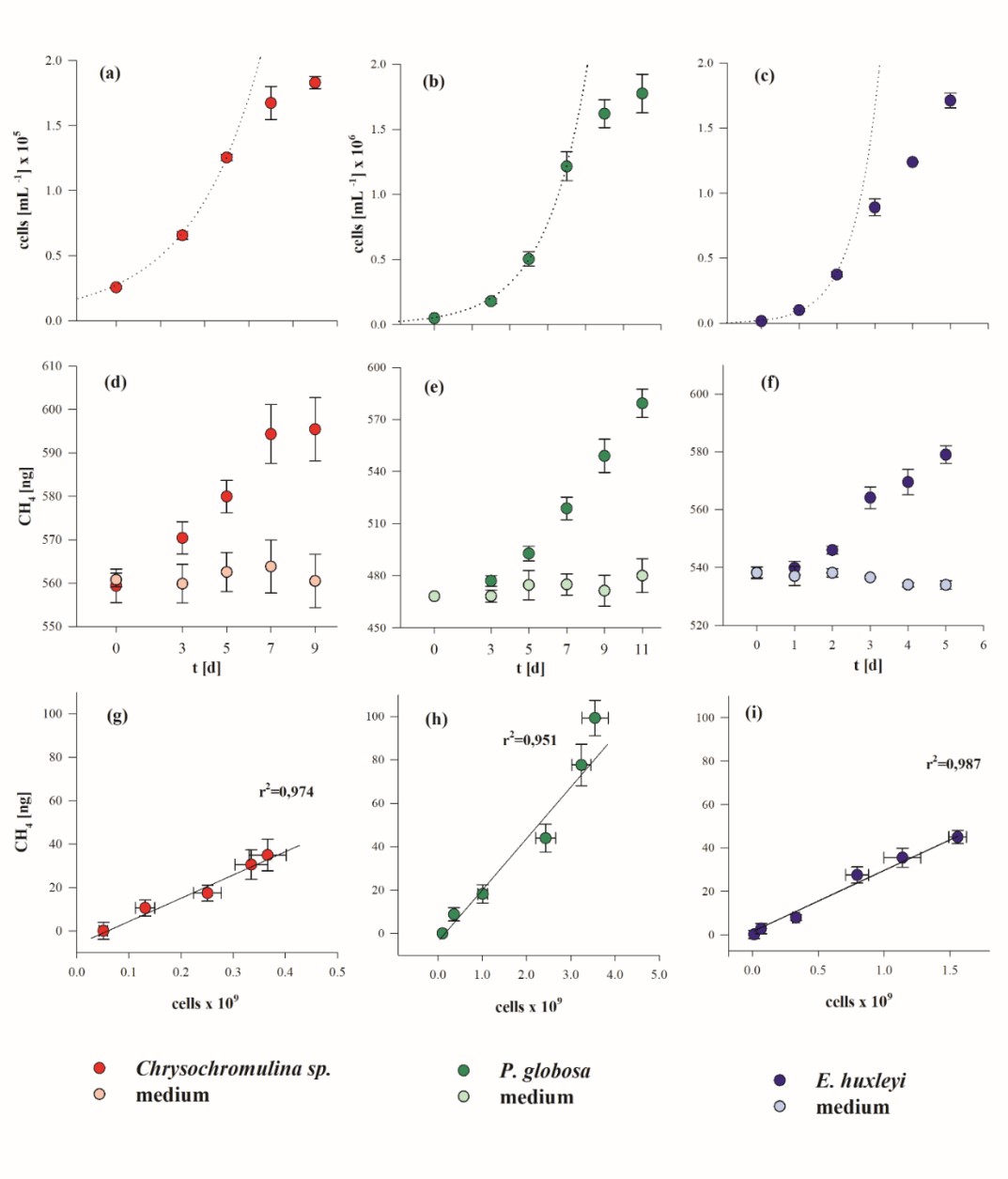


**Fig. 3: Cell growths (first panel), CH₄ production (middle panel) in course of time and correlation between the total number of cells and produced CH₄ (lower panel) from three algae species.** *Chrysochromulina sp.* **(left column a, d, g,),** *P. globosa* **(middle column b, e, h) and from** *E. huxleyi* **(right column c, f ,i). Please note that the cell numbers of** *Chrysochromulina sp.* **are presented in $10^5$ and P. globosa, E. huxleyi in $10^6$. Mean values of six (*Chrysochromulina sp., P. globosa*) and three (*E. huxleyi*) replicated culture experiments**
**are shown and error bars mark the SD.**

### 3.2 Stable carbon isotope values of CH$_4$ during incubation with $^{13}$C-hydrogen carbonate

Stable carbon isotope values of CH$_4$ ($\delta^{13}$CH$_4$ values) for *Chrysochromulina sp.* and *P. globosa* are presented in Fig. 4 (a, c).
We observed conversion of $^{13}$C carbon (provided by $^{13}$C-hydrogen carbonate) to $^{13}$CH$_4$ in cultures of both species, indicated

by increasing $\delta^{13}$CH$_4$ values over time. Stable isotope values increased from initial atmospheric (laboratory air) levels of -48.7
± 0.3 ‰ and -48.4 ± 0.10 ‰ up to +30.1 ± 10.2 ‰ and +245 ± 16 ‰ for *Chrysochromulina sp.* and *P. globosa,* respectively,
whilst the $\delta^{13}$CH$_4$ values of the control groups (algae without $^{13}$C-hydrogen carbonate or $^{13}$C-hydrogen carbonate in medium
without culture) did not change over time. The increase of $\delta^{13}$CH$_4$ values in the headspace-CH$_4$ depended on the amount of
released CH$_4$ that was added to the initial (atmospheric) background level. To calculate the $\delta^{13}$CH$_4$ values of the CH$_4$ source

which has raised CH$_4$ quantity above background level the Keeling-plot method (Keeling, 1958; Pataki et al., 2003) was used
(Fig. 4 b, d).

The calculated $\delta^{13}$CH$_4$ values of the CH$_4$ source were +1300 ± 245 ‰ (*Chrysochromulina sp.*) and +1511 ± 35 ‰ (*P. globosa*)
and thus close to the theoretical calculated $^{13}$C value of the DIC (2014 ± 331‰) resulting from the addition of $^{13}$C-hydrogen
carbonate. Please note that $^{13}$C-hydrogen carbonate stable isotope labelling experiment with *E. huxleyi* were already performed

by Lenhart et al. (2016) and were not repeated in this study. This is why $\delta^{13}$CH$_4$ values and the respective Keeling plot of *E.
huxleyi* are not shown in Fig. 4.

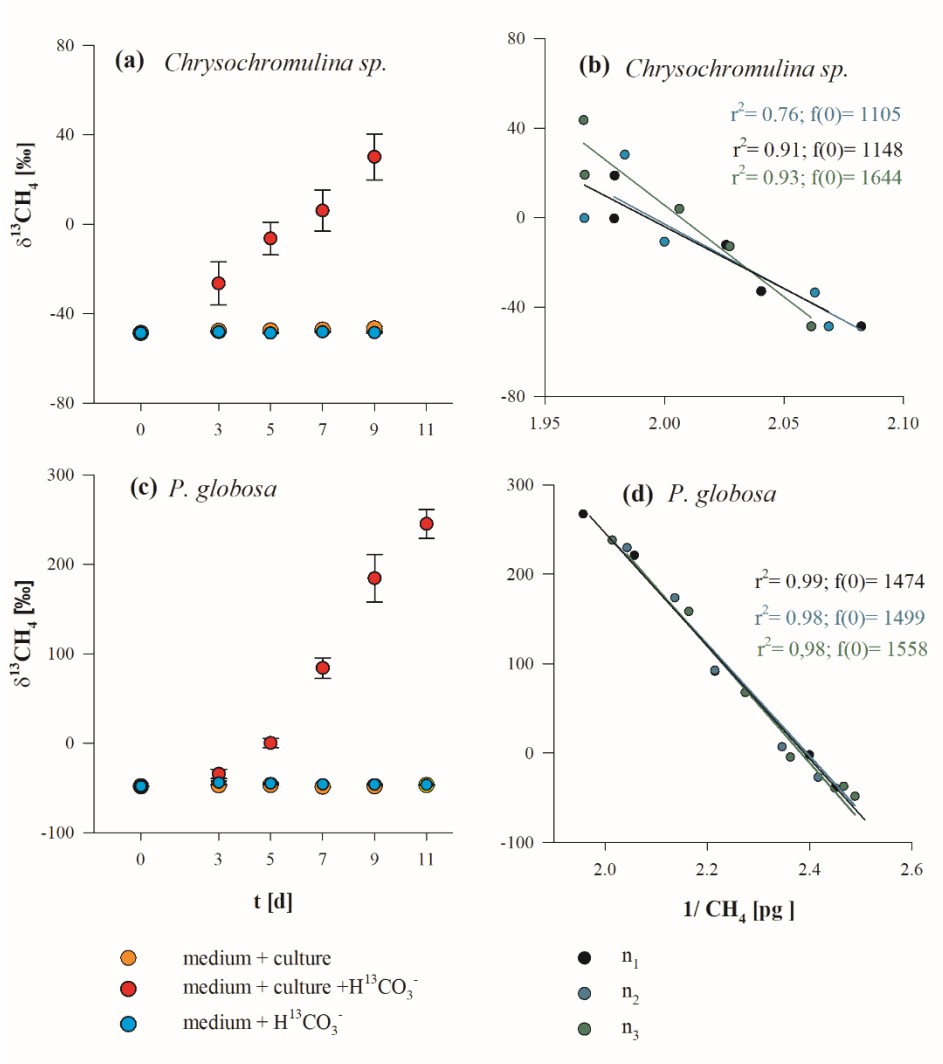

**Fig. 4: δ¹³CH₄ values (left column) and respective Keeling plots (right column) from *Chrysochromulina sp.* ( a,b) and *P. globosa* ( c,d) after the addition of H¹³CO₃⁻. The left column (a, c) shows the δ¹³CH₄ values of three investigation groups ("culture + H¹³CO₃⁻ ", "culture" and "H¹³CO₃⁻"), whereas each data point presented is the mean value of three replicated culture experiments with error bars showing SD. The right column shows the Keeling plots for the treatments "culture + H¹³CO₃⁻" from each replicated culture experiments (n₁, n₂, n₁) where f (0) refers to the ¹³C value of the CH₄ source.**

## 3.3 CH₄ production and production potential

Since the experiment in the section above (isotope measurements) was not designed to obtain POC quotas, we conducted an additional experiment to estimate CH₄ production rates of the three algal species.

From initial cell density of $22.5 \pm 3.1 \times 10^3$ cells mL$^{-1}$, $80.9 \pm 11.5 \times 10^3$ cells mL$^{-1}$ and $29.0 \pm 5.5 \times 10^3$ cells mL$^{-1}$ cultures were gown up to $37.0 \pm 9.2 \times 10^3$ cells mL$^{-1}$, $219 \pm 24.1 \times 10^3$ cells mL$^{-1}$ and $283 \pm 15.6 \times 10^3$ cells mL$^{-1}$ for *Chrysochromulina sp.*, *P. globosa* and *E. huxleyi,* respectively. These cell densities corresponded to the cell densities of exponential growth phase obtained from the experiment in section 3.1.

The POC normalized daily CH$_4$ production rate was highest in *E. huxleyi*, followed by *P. globosa*, and *Chrysochromulina sp.*. However, the cellular or POC normalized daily production rates of the three algal species were in the same order of magnitude (Table 1). We calculated the CH$_4$ production potential (CH$_4$PP), that is the amount of CH$_4$ produced within a week of growth (Gafar et al., 2018), to translate the cellular production rates ($\mu \times$ CH$_4$ cell$^{-1}$) of each species to community level. The CH$_4$PP was two order of magnitude higher for *E. huxleyi* than the other two species. This is a consequence of the higher growth rate of *E. huxleyi*.

We furthermore observed the oxygen concentrations during the light and dark periods to ensure oxic conditions. The measured oxygen concentrations were always saturated or supersaturated relative to equilibration with ambient air (Fig. S2).

**Table 1: Growth rate, cellular POC, CH$_4$ production rates and CH$_4$PP$_{(7days)}$ of *Chrysochromulina sp*. (n=4), *P. globosa* (n=4) and *E. huxleyi* (n=4). Values are the mean of four replicated culture experiments with SD.**

| | growth rate ($\mu$) | cellular POC | CH$_4$ production rate | | CH$_4$PP$_{(7days)}$ |
|---|---|---|---|---|---|
| | d$^{-1}$ | pg cell$^{-1}$ | ag CH$_4$ cell$^{-1}$ d$^{-1}$ | $\mu$g CH$_4$ g$^{-1}$ POC d$^{-1}$ | fg CH$_4$ |
| *Chrysochromulina sp.* | 0.21 $\pm$ 0.04 | 25.4 $\pm$ 4.0 | 44.5 $\pm$ 13.9 | 1.9 $\pm$ 0.6 | 1.0 $\pm$ 0.3 |
| *P. globosa* | 0.50 $\pm$ 0.06 | 7.0 $\pm$ 0.4 | 16.8 $\pm$ 6.5 | 2.4 $\pm$ 0.9 | 1.1 $\pm$ 0.4 |
| *E. huxleyi* | 1.09 $\pm$ 0.02 | 20.1 $\pm$ 0.7 | 62.3 $\pm$ 6.4 | 3.1 $\pm$ 0.4 | 121 $\pm$ 9.0 |

## 3.4 CH$_4$ formation from $^{13}$C labelled methyl thiol ethers

The three methylated sulphur compounds MSO, DMSO and DMS were tested for potential CH$_4$ formation in incubation experiments with *E. huxleyi*. The treatments were initiated in parallel from batch culture by inoculating $17.5 \pm 2.0 \times 10^3$ cells mL$^{-1}$ and cultures were grown to final cell densities of $1.77 \pm 0.08 \times 10^6$ cells mL$^{-1}$ (Fig. 5 a). Cell densities and CH$_4$ formation correlated in all treatments, while no difference in cell growth pattern or CH$_4$ formation was observed when isotope labelled methyl thioether and sulfoxides were added to the culture (Fig. 5 a, b, c). Differences between treatments were found in $\delta^{13}$CH$_4$ values of headspace CH$_4$. The initial $\delta^{13}$CH$_4$ value of headspace (-47.9 $\pm$ 0.1 ‰, laboratory air) increased slightly over time in untreated cultures (without isotope treatment) to -46.8 $\pm$ 0.3 ‰ (Fig 6.b).

In contrast, experiments where $^{13}C_2$-DMS, $^{13}C_2$-DMSO and $^{13}C$-MSO was applied to cultures of *E. huxleyi* $\delta^{13}CH_4$ values increased to $-31.0 \pm 1.1$ ‰, $-45.7 \pm 0.1$ ‰ and $+18.3 \pm 7.7$ ‰, respectively over a time period of 6 days (Fig. 6 a, b, c) and differed significantly from control groups ($p<0.05$).

The results unambiguously show that a fraction of the $^{13}C$-labelled methyl groups of the added substances was converted to 310 $^{13}C$-CH$_4$ in cultures of *E. huxleyi*. Much smaller changes in $\delta^{13}CH_4$ values were observed for controls of sterile filtered media where only $^{13}C_2$-DMS and $^{13}C$-MSO was added ($-42.8 \pm 1.7$ ‰ and $-43.9 \pm 0.2$ ‰ respectively, Fig. 6 a, c, day 6), whereas $\delta^{13}CH_4$ values did not change over time in the seawater controls (no addition of isotopic labelled compounds) and in the seawater controls treated with $^{13}C_2$-DMSO (Fig. 6 b). Based on the initial amount of $^{13}C$ label substance that were added to the cultures and the total amount of $^{13}CH_4$ at the end of the incubation period, $9.5 \pm 0.2$ pmol ($^{13}C_2$-DMS), $3,0 \pm 3,2$ pmol ($^{13}C_2$- 315 DMSO) and $30,1 \pm 3,6$ pmol ($^{13}C$-MSO) of 8.5 µmol were converted to CH$_4$.

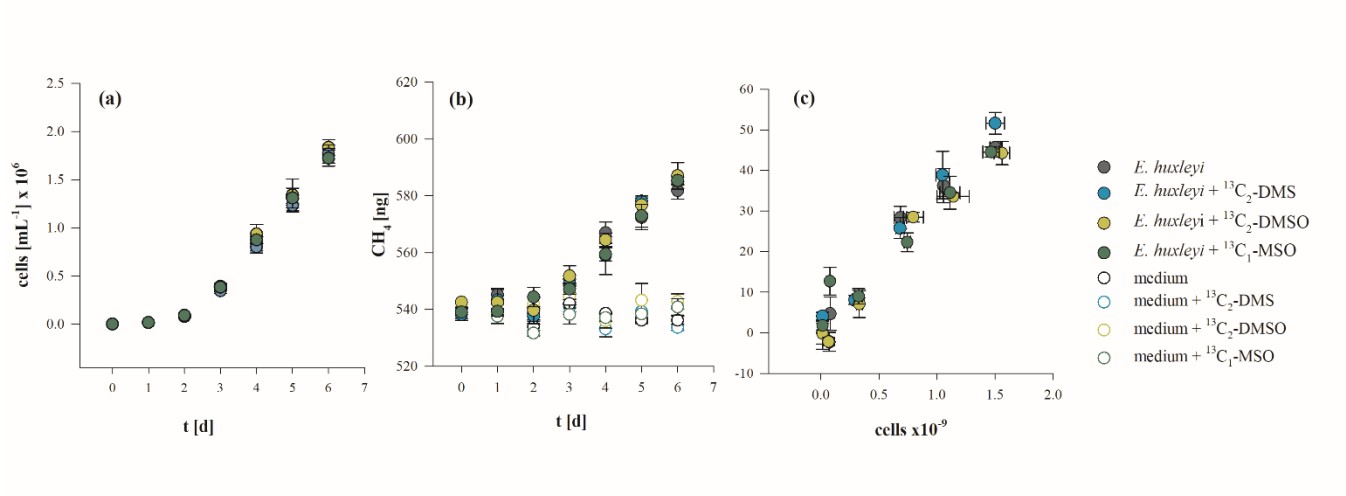

**Fig. 5 Cell growths (a), CH$_4$ production (b) and relation between the total number of cells and produced CH$_4$ (c) from *E. huxleyi* treated with $^{13}C_2$-DMS, $^{13}C_2$-DMSO and $^{13}C$-MSO or without any treatment. Mean values of three replicated culture experiments are shown and error bars mark the SD.**

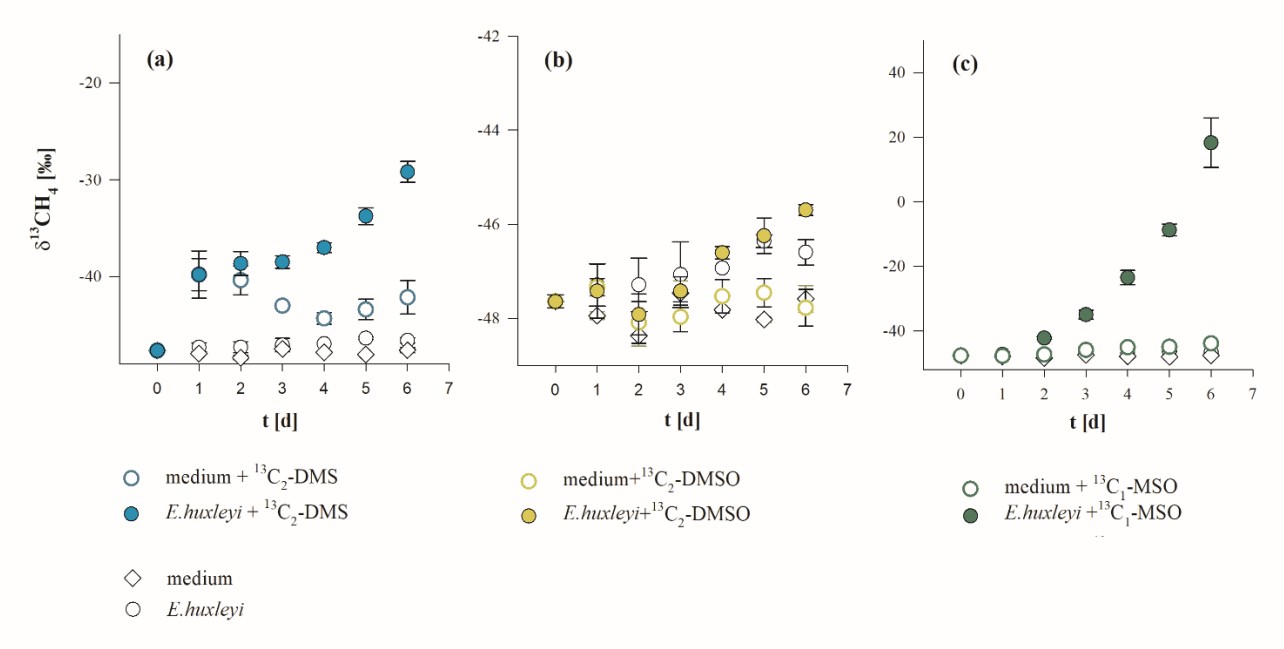

**Fig. 6** $^{13}$CH$_4$ values of headspace CH$_4$ in cultures of *E. huxleyi* supplemented with (a) $^{13}$C$_2$-DMS, (b) $^{13}$C$_2$-DMSO and (c) $^{13}$C-MSO. Mean values of three replicated culture experiments are shown and error bars mark the SD.

## 4. Discussion

Our results of CH$_4$ production and stable carbon isotope measurements provide unambiguous evidence that next to *E. huxleyi* (Lenhart et al., 2016) other widespread marine algal species namely *Chrysochromulina sp.* and *P. globosa* are involved in the production of CH$_4$ under oxic conditions at rates of $1.9 \pm 0.6$ to $3.1 \pm 0.4 \, \mu$g CH$_4$ g$^{-1}$ POC d$^{-1}$. The three investigated genera of marine phytoplankton have a world-wide distribution and they are representatives of the most widespread marine haptophytes (Schoemann et al., 2005; Thomsen, 1994; Brown and Yoder, 1994). The results indicate that CH$_4$ production could be a common process across marine haptophytes. We first discuss the stable isotopic evidence of CH$_4$ formation, the role of precursor compounds and likely mechanisms involved. Finally, we discuss the laboratory CH$_4$ production rates in relation to its potential significance in marine environments and provide a first rough estimation how these production rates might contribute to CH$_4$ concentration in oxic surface waters previously reported in open ocean algal blooms.

In cultures of *Chrysochromulina sp.* and *P. globosa*, that were treated with $^{13}$C-labelled hydrogen carbonate, $\delta^{13}$CH$_4$ values increased with incubation time, clearly resulting from the conversion of $^{13}$C-hydrogen carbonate to $^{13}$CH$_4$. These results demonstrate that all three investigated algal species are instrumental in the production of CH$_4$ under oxic conditions (Fig. S1) and that hydrogen carbonate serves as a carbon source for $^{13}$CH$_4$. Our findings are in agreement with the stable isotope evidence of CH$_4$ production by *E. huxleyi* (Lenhart et al., 2016). However, we do not consider hydrogen carbonate as the direct carbon

precursor of CH4. In a first step hydrogen carbonate and its isotope label is converted to $CO_2$ and subsequently fixed by algal primary production forming POC. Therefore, we would expect a large fraction of the $^{13}C$ label of the hydrogen carbonate (+2014 ± 331‰) to be transferred to the POC towards the end of the experiment (where the volume normalized POC content is highest). The experiments were started by inoculation of cells from pre-cultures, that were grown on DIC with natural $^{13}C/^{12}C$ abundance ($\delta^{13}C$ values ~0 ‰). This means that during ongoing incubation the $\delta^{13}C$-POC value should get close to $\delta^{13}C$-DIC values, resulting from the addition of $^{13}C$-hydrogen carbonate, when cultures grow in the new $^{13}C$ enriched medium. Consequently, the $\delta^{13}C$-POC values are considered to be somewhat lower than the theoretically calculated $\delta^{13}C$-DIC values (+2014 ± 331‰) of the medium. Our assumptions are in line with the $\delta^{13}CH_4$ source signature values (averaged over 9 or 11 days respectively), obtained via Keeling plot method, which were +1300 ± 245 ‰ and +1511 ± 35 ‰ for *Chrysochromulina. sp*. and *P. globosa*, respectively and thus were somewhat lower than for the theoretical calculated $^{13}C$ value of the DIC (+2014 ± 331‰) resulting from the addition of $^{13}C$-hydrogen carbonate. Unfortunately, $\delta^{13}C$-DIC and $\delta^{13}C$-POC values could not be determined in our set of experiments to allow more detailed calculations. However, our results clearly indicate that hydrogen carbonate is the principle inorganic carbon precursor of $^{13}CH_4$ produced in algae, but intermediate metabolites are likely to be formed from which CH4 is released, possibly by cleavage of sulphur-bonded methyl groups of methyl thioethers and sulfoxides (Althoff et al., 2014; Lenhart et al., 2016; Benzing et al., 2017).

## 4.1 CH4 formation from $^{13}C$ labelled methyl thioethers

*Methyl thioethers are precursors of CH4*

Methyl thioethers and their sulphoxides are ubiquitous in marine environments as they are often produced by algae at substantial rates. It is also known that these compounds are metabolized in the three investigated algal species (Liss et al., 1994; Keller, 1989). Based on the addition of $^{13}C_2$-DMSO, $^{13}C_2$-DMS and $^{13}C$-MSO, where only the sulphur-bonded methyl groups (–S-CH3) were 99% labelled with $^{13}C$, it was possible to clearly monitor $^{13}CH_4$ formation by stable carbon isotope measurements in cultures of *E. huxleyi*. The $\delta^{13}CH_4$ values, increased over time significantly in $^{13}C_2$-DMS, $^{13}C_2$-DMSO and $^{13}C$-MSO treated cultures, above the $\delta^{13}CH_4$ values of the control groups (Fig. 6 a-c). The $^{13}C$-labelling experiment showed that DMS, DMSO, and MSO are potentially important methyl-precursors for CH4 but the contribution of these compounds to the overall CH4 production in cultures of *E. huxleyi* could not be determined in our experiments due to the complexity of the formation of these compounds in the algal cells. This can be illustrated by the following. The contribution of a substance to the total CH4 released is the product of both the added $^{13}C$-labeled fraction (added to the waters sample and uptake by the cells) and the internally formed fraction of these compounds (DMS, DMSO and MSO) which will roughly show natural $^{13}C$ abundance. Therefore the stable isotope value of CH4 will be diluted by the fraction of naturally formed methyl sulfur compounds in the algal cells and thus the contribution of DMS, DMSO, and MSO to CH4 formation can therefore not be estimated on the basis of their added amount alone. The $^{13}CH_4$ quantity from conversion of added $^{13}C$ labelled substance

contributed 0.03% ($^{13}C_2$-DMSO) up to 0.84% ($^{13}$C-MSO) to overall released $CH_4$. However, even if the added $^{13}$C labelled compounds might only explain ≤ 1% of $CH_4$ formed by the algae their overall contribution (including non-labelled sulfur compounds which we are not able to measure) might provide a much larger fraction of the released $CH_4$. The intracellular DMS concentration can reach 1 mM (Sunda et al., 2002) in cells of *E. huxleyi*, while the concentration of added $^{13}C_2$-DMS was 0.01 mM in medium (final concentration). If intracellular $^{13}C_2$-DMS was in equilibrium with bulk seawater $^{13}C_2$-DMS and

all $CH_4$ would be produced from intracellular DMS, then the contribution of the $^{13}$C labelled compound would be about 1%. However, even if the biggest fraction of $CH_4$ in algae cultures was not released by the $^{13}$C labelled substances, the significant increase in delta notation in $^{13}C_2$-DMS, $^{13}C_2$-DMSO and $^{13}$C-MSO treated cultures above the $\delta^{13}CH_4$ values of the control groups demonstrate that $^{13}$C labelled precursor substances were converted to $CH_4$ by algal cultures (Fig. 6 a-c). This is also indicated, when the absolute conversion quantities of $^{13}$C-labelled substance in algal cultures are considered: these were ca.

nine ($^{13}C_2$-DMS), three ($^{13}C_2$-DMSO) and thirty ($^{13}$C-MSO) times higher than in seawater control groups. Hence, the stable isotope labelling approach should be considered as a proof of concept, that methyl groups of all tested substance serve as precursor compounds of $CH_4$.

These isotope labelling results are also in good agreement with recent results from laboratory experiments where $^{13}$C-MET was added to cultures of *E. huxleyi* (Lenhart et al., 2016). In addition, we also found an indication for a purely chemical $CH_4$

formation pathway from control samples using sterile seawater and addition of either $^{13}C_2$-DMS and $^{13}$C-MSO. The $^{13}C_2$-DMS spiked seawater group and the $^{13}C_2$-DMS spiked algae group are very close to each other up to day 2 (see Fig. 5a and Fig. 6a). For this time period, it can be assumed that the chemical conversion has taken place in both samples to the same extent, since the samples are relatively similar, because the algal cell density is only 5% (day 2) of the final cell density. However, the following days (day 3 to day 6), when algal cell density increased drastically, the $\delta^{13}CH_4$ values of the algae cultures also

increased significantly compared with $\delta^{13}CH_4$ values of the seawater. This clearly indicates that conversion of $^{13}C_2$-DMS to $CH_4$ increases with increasing cell counts.

However, the relatively slight increase in $\delta^{13}CH_4$ values in the control samples (Fig. 6 a, c) implicates that this is only a minor pathway. The $CH_4$ conversion from $^{13}$C-DMS and $^{13}$C-MSO in seawater was approximately 3- and 30-fold lower than in the corresponding treatments with algae and becomes only obvious when applying very sensitive stable isotope labelling

experiments. A similar observation was already made by Lenhart et al. (2016) when applying $^{13}$C-MET in seawater. However, this observation might be in agreement with previously findings by Zhang et al. (2015), who described a photochemically and CDOM related conversion of DMS to $CH_4$ in oxygenated natural seawater.

*Potential mechanism of $CH_4$ formation from thioethers*

The $CH_4$ formation from thioethers (MET, DMS) and their corresponding sulphoxides (MSO, DMSO) might be catalysed by nonheme oxo iron (IV), thus forming methyl radicals (·$CH_3$) from homolytically broken sulphur methyl bounds (R-$CH_3$) leading to $CH_4$ under oxidative conditions (Althoff et al., 2014; Benzing et al., 2017). The tested compounds are found in high cellular concentrations in *E. huxleyi*, *Chrysochromulina sp.* and *P. globosa* and non heme oxo iron (IV) have been identified

as active intermediates in the catalytic cycles of a number of biological enzymatic systems (Hohenberger et al., 2012).

Therefore, the postulated reaction might be a likely pathway for $CH_4$ production in investigated alga species. Furthermore, DMS and DMSO were described to be part of an antioxidant system as these compounds can readily scavenge hydroxyl radicals in cells of *E. huxleyi* (Sunda et al., 2002). Furthermore, $CH_4$ is released via a methyl radical, that is subtracted from DMSO when hydroxyl radicals being scavenged – and accordingly DMS after its sulphoxidation (Herscu-Kluska et al., 2008). Since MET and MSO have similar functional groups to DMS and DMSO respectively, it was proposed that the reaction

described above is taking place analogously for these compounds (Bruhn et al., 2012; Lenhart et al., 2015a). Consequently, the $CH_4$ formation in investigated algal species might be a response of oxidative stress, that forms hydroxyl radicals or other reactive oxygen species (ROS), which in turn might react with the applied methylated sulphur compounds generating methyl radicals and eventually $CH_4$.

The algal metabolites DMSP, DMS and DMSO are ubiquitous in marine surface layers and nanomolar concentrations were

found in blooms of *Chrysochromulina sp.*, *P. globosa* and *E. huxleyi*. Several field studies showed that these compounds are linked to $CH_4$ formation in seawater (Zindler et al., 2013; Damm et al., 2008; Florez-Leiva et al., 2013). The authors proposed that DMSP and their degradation products DMSO and DMS are used by methylotrophic methanogenic archaea, inhabiting anoxic microsites, as substrates for methanogenesis. In addition it was reported that, if nitrogen is limited but phosphorus is replete, marine bacteria might also use DMSP as a carbon source, thereby releasing $CH_4$ (Damm et al., 2010).

One scenario which we cannot rule out would be a production of $CH_4$ precursors by algae and a usage of these precursors by bacteria to produce $CH_4$. While we think that this is less likely than $CH_4$ production by algae alone, it would, if true, show that bacteria need algae-produced precursors to produce $CH_4$. The latter scenario would be relevant in the field because algae co-exist with bacteria in the oceans. Therefore bacteria might be involved in the $CH_4$ production process in our cultures, but even if they were they still would depend on algal growth. For further discussion of a potential contribution of heterotrophs and/or

methanogenic archaea see supplementary material (S3). The correlations we describe in the supplementary material clearly show that $CH_4$ production depends on algal growth. It is therefore highly unlikely that bacteria are solely responsible for $CH_4$ production in our cultures.

## 4.2 POC normalized production

For all three algal species significant correlations between $CH_4$ mass and cell density was found ($r^2 > 0.95$ for all species, Fig 3 g, h, i), suggesting that $CH_4$ formation occurred over the entire growth curve.

However, since $CH_4$ production can only be determined in the exponential phase (Langer et al., 2013) we additionally ran dilute batch cultures to determine $CH_4$ production. All three species displayed similar $CH_4$ production ranging from $1.9 \pm 0.6$ to $3.1 \pm 0.4\ \mu g\ CH_4\ g^{-1}\ POC\ d^{-1}$ with *Chrysochromulina sp.* and *E. huxleyi* showing the lowest and highest rates, respectively.

The $CH_4$ production for *E. huxleyi* was found to be twofold higher than rates reported for the same strain and comparable

culture conditions by Lenhart et al. (2016) (0.7 µg $CH_4$ $g^{-1}$ POC $d^{-1}$). The lower production reported by Lenhart et al. (2016) may be explained by the fact that $CH_4$ production was not obtained from exponentially growing cultures. We also compared the cellular $CH_4$ production rates of *E. huxleyi* reported by Scranton (1977) with those of our study. Scranton (1977) reported a production rate of $2 \times 10^{-10}$ nmol $CH_4$ $cell^{-1}$ $hr^{-1}$. This value is close to the production rate of $1.6 \times 10^{-10}$ nmol $CH_4$ $cell^{-1}$ $hr^{-1}$

in our study. Scranton (1977) concluded from observed $CH_4$ production rates in laboratory experiments that natural populations might be adequate to support the widespread supersaturations of $CH_4$ observed in the open ocean. However, we do suggest that $CH_4$ production of various algae might differ substantially under changing environmental conditions, as already shown for terrestrial plants (Abdulmajeed and Qaderi, 2017; Martel and Qaderi, 2017). Moreover, the cellular concentrations of potential precursor compounds such as methylated sulphur compounds might vary greatly between species and cultures. The

investigated algal species can reach millimolar intracellular concentrations of DMS and DMSP (Sunda et al., 2002; Liss et al., 1994; Keller, 1989) and even if the conversion rate of methylated sulphur compounds to $CH_4$ in algal cells might be low, they could be sufficient to explain a substantial fraction of the $CH_4$ production rates by marine algae.

### 4.3 Implication for the marine environment and algal blooms

In general, the distribution of chlorophyll has not shown a consistent correlation with $CH_4$ distributions in field studies. There are studies in which no correlation was observed (e.g. Lamontagne et al., 1975; Foster et al., 2006; Watanabe et al., 1995) or a correlation was found within a few depth profiles (Burke et al., 1983; Brooks et al., 1981). Many field measurements in oxygenated surface waters in marine and limnic environments have shown examples of elevated $CH_4$ concentrations spatially related to phytoplankton occurrence (e.g. Conrad and Seifer, 1988; Owens et al., 1991; Oudot et al., 2002; Damm et al., 2008;

Grossart, et al., 2011; Weller et al., 2013; Zindler et al., 2013; Tang et al., 2014; Bogard et al., 2014; Rakowski et al., 2015). Taken together these studies suggest that phytoplankton is not the sole source of $CH_4$ in oxygenated surface waters, but importantly they also suggest that phytoplankton is one of the sources of $CH_4$. We therefore compared the $CH_4$ production rates of our cultures with two field studies for the Pacific Ocean (Weller et al., 2013) and the Baltic Sea (Schmale et al., 2018) to evaluate the potential relevance of algal $CH_4$ production. It was estimated that the gross $CH_4$ production in a southwest

Pacific Ocean mesoscale eddy is 40 - 58 pmol $CH_4$ $L^{-1}$ $d^{-1}$ (Weller et al., 2013). Using reported phytoplankton cell densities ($1.7 \times 10^8$ to $2.9 \times 10^8$ cells $L^{-1}$, Weller et al., 2013), we calculated a maximal cellular production of 5.5 ag $CH_4$ $cell^{-1}$ $d^{-1}$ for this eddy. The species investigated in this study showed ca. 3-11 times higher cellular production (Table 1). Hence each of the three haptophyte algae studied here could account for the $CH_4$ production reported by Weller et al. (2013).

Schmale et al. (2018) reported $CH_4$ enrichments that were observed during summer in the upper water column of the Gotland

Basin, central Baltic Sea. Furthermore they found that zooplankton is one but not the only $CH_4$ source in the oxygenated upper waters. While the authors ruled out a major contribution of algae to the observed sub-thermocline $CH_4$ enrichment because of the low sub-thermocline phytoplankton biomass, they considered a primary production associated $CH_4$ formation as one likely

source in the phytoplankton-rich mixed layer. The average phytoplankton carbon biomass of the mixed layer was approximately 600 µg L$^{-1}$ (averaged from Fig. 9 in Schmale et al., 2018). For the reported average net CH$_4$ production rate in the mixed layer (95 pmol CH$_4$ L$^{-1}$ d$^{-1}$), we calculated that a production rate of 2.5 µg g$^{-1}$ POC d$^{-1}$ is required if the CH$_4$ is produced by the algal biomass. This rate would be within the range of CH$_4$ production rates observed in our study. These calculations should be considered as a first rough estimate to assess whether CH$_4$ production rates of laboratory grown cultures can significantly contribute to CH$_4$ supersaturation associated with phytoplankton. We did not distinguish between species and did not take into account environmental factors or the complexity of microbiological communities.

Judging from cellular production, the species studied here are of similar importance for oceanic CH$_4$ production in biogeochemical terms. Regarding the highest cellular production, that of *E. huxleyi* as 100%, *P. globosa* produces 27% and *Chrysochromulina sp.* 71% (Table 1). However, several recent studies have emphasized that the production potential (PP), as opposed to cellular production,  is a biogeochemically meaningful parameter (Gafar et al., 2018; Marra, 2002; Schlüter et al., 2014; Kottmeier et al., 2016). The concept of the production potential goes back at least to the first half of the 20$^{th}$ century (Clarke et al., 1946). Briefly, the production potential of substance X is the amount of X which a phytoplankton community or culture produces in a given time. For details see Material and Methods and references above. The cellular production by contrast is the rate of production of X of a single cell, and therefore the cellular production is ill qualified to express community-level production.

We calculated the CH$_4$-PP (Material and Methods) for our three species, and when the one of *E. huxleyi* is considered 100%, *P. globosa* has a CH$_4$-PP of 0.9%, and *Chrysochromulina sp.* 0.8% (Table 1). In terms of CH$_4$ production in the field, therefore, *E. huxleyi* out-performs the other two haptophytes by two orders of magnitude. It can be concluded that the CH$_4$-PP under given environmental conditions is species-specific and therefore community composition will have an influence on algal sea surface water CH$_4$ production.

It can be hypothesized that changing environmental conditions might drastically affect algal CH$_4$ production, which has to be taken into account when calculating annual averages. The effect of dominant environmental parameters such as light intensity and temperature on algal CH$_4$ production will therefore be the subject matter of future studies.

**Competing interests:** The authors declare that they have no conflict of interest.

**Acknowledgment**: We thank Markus Greule, Bernd Knape and Stefan Rheinberger for conducting analytical measurement and technical support that helped to produce this dataset. F.K. and T.K. were supported by the German Research Foundation (DFG; KE 884/8-2, KE 884/11-1 and KE 884/16-2), G.L. by the Natural Environment Research Council (NE/N011708/1).

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
