# Peer review of "Methane production by three widespread marine phytoplankton species: release rates, precursor compounds, and potential relevance for the environment"

_Biogeosciences, 2019_

## Referee Comment (RC1) · Mary Scranton (Referee) · 17 Jul 2019

This paper presents an interesting discussion about the importance of methane production by several species of algae under aerobic conditions in the ocean. The authors’ experiments are original and convincing but I think they overstate (or ignore) the extent to which this process can result in methane excess concentrations in open ocean surface water. In turn the minor role of that excess production to the atmospheric methane budget is not clearly explained. Below are some substantive criticisms and some minor corrections.

[Figure]

Line 17: The abstract indicates that the importance of oceanic methane production to the global methane budget is unknown but this is not discussed further in article and is misleading in any case since the ocean is known to be a very small contributor to the atmosphere. I am tired of proposals and papers that use the atmospheric methane budget to justify all studies of basic methane geochemistry. Surely it is enough to note a widespread and unexplained phenomenon which one is trying to explain mechanistically. I suggest adding a sentence or two to the introduction indicating why you are bothering to do this study and de-emphasizing how it might affect global methane budget. You are better off being straightforward and admitting that the real question is that methane is known to be produced in the oxic oceanic mixed layer and after more than 40 years no one really understands why. Give some idea of what actual flux of methane to atmosphere from ocean is thought to be. This HAS been calculated a number of times.

Line 98: Were cultures axenic? How was this determined? Sterile technique is not enough if bacteria are intrinsic to algal cultures. Bob Guillard told me this when I was using his culture collection. I personally don't think that there are anaerobic bacteria producing methane in rapidly photosynthesizing cultures, but one should be accurate.

Line 115: When calculating the amount of methane produced, was fraction dissolved included? With a large headspace, this may be small but should be mentioned. Were samples equilibrated with headspace before methane measured? The authors mention that oxygen was sometimes supersaturated, but was this relative to headspace or equilibration with ambient air?

Line 133: Concentrations (final) of added substrates should be given for comparison with natural concentrations. If possible give concentrations of these substrates in medium at start of incubation with and without addition of substrate.

Line 327: If the labelled methyl groups yield only a small percentage (less than 1%) of total methane produced where is the other methane coming from? Is this result

consistent with field observations that show only a weak link if any between DMS or DMSO and excess methane in surface water? This point needs more elaboration since the question of the source of excess methane in seawater has been plagued by studies that show methane can be produced by a process but that rates are far lower than are needed to explain natural surface water values. Here is where a link to ambient DMS, DMSO or MSO concentrations should be made. I think this point is a key issue.

Line 400: Weller et al may have found a correlation between chlorophyll a and methane concentrations but there were many studies in the older literature (1970s and 80s) where no such correlation was observed. I recommend authors go back and read over some of these earlier papers and confirm that measured production rates from this study can support other previously observed methane fluxes. Also see thesis by Scranton (1977) where methane production was examined in cultures by several species including Emiliani huxleyi (called Coccolithus huxleyi in my thesis) and T. pseudonana. I observed methane production in a much less sophisticated experimental setup and concluded that natural populations of the algae I studied might be adequate to support the widespread supersaturations of methane seen in the open ocean (including in places where no dense algal blooms were observed). Perhaps your results can be compared to mine or to other studies that report cell abundances and air-sea fluxes. A citation to a downloadable copy of my thesis is below. Scranton MI (1977) The marine geochemistry of methane. Citable URI https://hdl.handle.net/1912/1616. DOI10.1575/1912/1616

Minor issues: Equation 7: There should be a factor of 1000 to convert ratios to per mille values Figure 1: Plot control values here too. Line 268: Should it be "were applied" not "where applied"? Line 308: Inoculation OF cells?

---

## Referee Comment (RC2) · Anonymous Referee #2 · 24 Jul 2019

The present paper presents an interesting study about methane production under oxic conditions in marine environments. This so called "methane paradox" is a very important research field to understand methane emissions from oceans (and lakes) and has recently received strong interest by a number of investigators from different scientific disciplines. The author presents data from incubation experiments conducted with three different algal species. Methane production rates were determined with different methanogenic substrates (13C-labled) using a stable isotope approach. A similar kind of studies was previously conducted for Emiliania huxleyi by Lenhart et al. (2016)

and the isotope approach was successfully used in diverse investigations by Frank Keppler before to examine terrestrial methane production. The novel outcome in the present study is (1) that also other widespread haptophytes have the potential to produce methane under anoxic conditions; and (2) methylated sulphur compounds (e.g. DMS), that are known to be enriched in the investigated algae species, present potential substrates. In addition, the authors present an attempt to transfer their results to an algal bloom in the Pacific Ocean to discuss the potential relevance of algal methane production.

The experiments are well thought out and the results present an additional piece in the complex puzzle. There are lots of little corrections needed and from my point of view some sentences need another structure to make the content more accessible for readers that are not familiar with the topic in detail (especially in the method section, e.g. PP, exponential growth rate). I will give a few examples below.

Some minor and major points need to be addressed and I therefor recommend a publication after major revision.

Major issues:

Line 95ff. The experimental design is very complex. A flowchart for the method section would be helpful for the reader.

Line 98ff. How clean are the algal culture samples (purity)? Small differences in the degrees of contamination with archaea/bacteria (nitrogen limited bacteria, Line 69, Damm) between the cultures may have an impact on CH4 production rate. Does the web link give information about the purity of the culture?

Line 133. What is the difference in concentration of NaHCO3 between natural and inoculated water sample? Why did the authors added this amount of tracer? Should be mentioned.

Line 137. Can you explain if aggregates or sediment was visible in the incubation?

[Figure]

Line 141. Here, you should explain in more detail why an exponential growth rate is important to best compare CH4 formation between the experiments. From this sentence one could assume that Langer performed already methane production rate experiments with algae that indicated that exponential growth rates are important. From my point of view the activity of the cell is important for the turnover of these substrates and not their reproduction.

It should be mentioned in the method or result section that microbial methane turnover takes place in the incubations and the production rates presented are minimum rates > because methane oxidation is not considered in the calculations (e.g. see methods in de Angelis and Lee).

Line 172. Why was exactly this amount of substrate (DMS. . .) injected and is this comparable with natural environments (concentrations). Substrate concentrations definitely affect the turnover and the addition of tracers/substrates should not impact the sample too drastically. Why did the authors did not applied MET (and DMSP) as a precursor that was tested before successfully by Lenhart et al.?

Line 327ff. Is it possible that a natural microbial community is needed for the turnover of these substances to methane? If the incubations are without contaminations (sterile filtered seawater, pure culture), the production rates might be low because of the missing community. The algae may only provide the precursors. Might be a point that could be discussed here.

Line 335. I have a different impression. Figure 4a: At day 2 the d13C values are very close to each other. In Figure 4b all values from beginning to the end of the incubation time are very similar. Only Figure 4c shows a clear difference between culture and control over the course of the experiment. Add in the figure caption that also controls are plotted, not only results from cultures.

Line 381. Argumentation is difficult. Only because Lenhart could prove a contamination-free incubations, this result cannot be transferred to all the incubation

that will be performed by the working group afterwards. See comments/concerns to this topic above. Since the argumentation is difficult to follow, I suggest to discuss this topic less dominant and integrate this part somewhere else (not under a separate title). Also 50% of the text is nearly copied from the introduction (doubling!).

Chapter 4.3 I would recommend to perform an additional calculation to show that algal CH4 production is an important mechanism that can explain air/sea methane fluxes and methane enrichments. For example Schmale et al. (2018) gives detailed data about phytoplankton biomass (e.g. Prymnesiales) and production rates needed to maintain air/sea fluxes and subthermocline methane enrichments. There are probably also other papers available that could be used for such calculation.

Minor issues:

Title: I recommend writing "potential relevance for the environment". A direct transfer of laboratory studies/results into field observations is difficult.

Line 24. Please also give the productions rates per cell in the abstract. Temperature is not needed to be mentioned in the abstract.

Line 27ff. It should be mentioned here that the conversion of methylated sulphur compounds to methane was only responsible for less than 1% of the observed methane production (line 327ff).

Line 26-29. The word "clearly" is used to often.

Line 30. "Relevance for the environment" is one major issue in the title but is reduced here to a little sentence. This part should be extended.

Line 49. How can "emissions from freshwater" explain the CH4 concentration in ocean surface water? Line 50. Shorten the sentence and delete "that has been often...". "Well-known" means "often reported"

Line 55-58. This paragraph should be moved to line 46. It might be better to start with

this overall review before listing the recent specific studies to oxic methane production in lakes and ocean.

Line 60. May also mention Valle and Karl (2014) who used in situ MPn concentration in a 14C approach and showed that dissolved MPn in surface waters cannot account for methane oversaturation.

Line 98. A bracket is missing (RC...). Is it clear for the reader for what the web link is good for?

Line 102. Delete "in" in front of "natural"

Line 110ff. Why did the authors used different volumes (medium and headspace)?

Line 119. What is meant with "main cultures"? Is this the investigated culture in the incubation?

Line 119ff. I would suggest to transfer the cell densities to the result section (3.1).

Line 122. E. huxleyi was sampled daily! What do you mean with overall sampling interval: 9,11,6 days? Why this odd order? And why did you sample the cultures in different intervals? If there is a reason for that it should be explained.

Line 128. Why only three data points for E. huxleyi. From Figure 1c it seems to be plausible to use four.

Line 131. It is always worth to have a repetition to support the previous results.

Line 134. The delta is missing in d13C

Line 135. Suggest to write: "...values of the methane precursor..."

Line 144. "...measured at the end..."

Line 145. Suggest to write "For this additional experiment..."

Line 146. Suggest putting the cell densities in the result section (see above).

Line 153. "ag" is the abbreviation for what?

Line 158. The program "Image J" is produced by which company?

Line 174. I still think that cell densities should be implemented in the result section (see above).

Line 176. The target/design of the experiment in section 3.2 is still unknown!

Line 189. Analyzed. See line 180

Line 193. Write: "... (based on three...)".

Line 204. Delete: "...at a temperature...". Here and in some other parts of the result section you mention details that were mentioned before in the method section.

I would start with a sentence that makes clear that you are talking about the incubation with 13C-labelled hydrogen carbon (2.3).

Line 208. Also here delete the repeated information: "These rates were obtained...". Check the entire result section to avoid redundancy.

Line 211. The cell density should only be mentioned here and not in the method section!

Line 214. Where is the control group plotted?

Figure 2. Black and blue dots are difficult to distinguish. Even if it is "only" the control sample – make the visibility easier. The x-axis should be 1/CH4. Right?

Line 249. See above. Not clear why the exponential growth phase is important and not the cell activity.

Line 251. The equation is already described above (2 and 4). Avoid doubling. See comment above.

Line 254. The sentence should end with (Tab. 1).

Line 256. "community level" sounds odd in this context. May you can find a better description.

Line 266. It starts again with information that was mentioned before in the methods.

Line 271. The sentence should end with "(Fig. 4)".

Figure 4b. Change the x-axis to 13C2 (add 2).

Figure 4c. Change the x-axis to MSO (not MES, see caption).

Line 279. Add the control sample in the Figure.

Line 302. In the present study only the turnover of 13C-hydrogen carbonate by two algal species was investigate. Lenhart applied the isotope technique for E. huxleyi.

Line 307. (with highest cell numbers) is out of context. Please rephrase.

Line 333. In future investigations I would suggest a dark incubation to exclude methane production by UV or visible light (line 70ff).

Line 352ff. Did Althoff really proved that the "reactivity" is the driving force in her experiments? Or are point 1 (label concentration) and 2 (penetration) also possible explanations for her observations?

Line 360ff. Sentence too complex. Devide in two parts.

Line 363 and 365. Too often "furthermore". Rephrase.

Line 391. Interesting. But it needs be explained in more detail why the growth rate impacts the methane production. See above.

Line 411. Include/explain why PP is meaningful parameter.

---

## Author Comment (AC2) · 14 Sep 2019

Point-by-point response to the issues raised by referee#1 (Mary Scranton)

We thank the referee for the constructive comments and suggestions which have helped to improve the manuscript.

Referee #1: This paper presents an interesting discussion about the importance of methane production by several species of algae under aerobic conditions in the ocean. The authors' experiments are original and convincing but I think they overstate (or

ignore) the extent to which this process can result in methane excess concentrations in open ocean surface water. In turn the minor role of that excess production to the atmospheric methane budget is not clearly explained. Below are some substantive criticisms and some minor corrections.

Authors: We appreciate the positive evaluation of our manuscript. The criticisms are addressed and corrections are made below.

Referee #1 Line 17: The abstract indicates that the importance of oceanic methane production to the global methane budget is unknown but this is not discussed further in article and is misleading in any case since the ocean is known to be a very small contributor to the atmosphere. I am tired of proposals and papers that use the atmospheric methane budget to justify all studies of basic methane geochemistry. Surely it is enough to note a widespread and unexplained phenomenon which one is trying to explain mechanistically. I suggest adding a sentence or two to the introduction indicating why you are bothering to do this study and de-emphasizing how it might affect global methane budget. You are better off being straightforward and admitting that the real question is that methane is known to be produced in the oxic oceanic mixed layer and after more than 40 years no one really understands why. Give some idea of what actual flux of methane to atmosphere from ocean is thought to be. This HAS been calculated a number of times.

Authors: We agree with the referee and thus have modified the Abstract and Introduction. The first sentence of the abstract now reads: "Methane ($CH_4$) production within the oceanic mixed layer is a widespread phenomenon, while the underlying $CH_4$ producing mechanism is still topic of scientific debate" We further added two sentences to the introduction: "The world's oceans are considered to be a minor source of methane ($CH_4$) to the atmosphere (1-3 %, Saunois et al., 2016). However, in recent years the widespread occurrence of in situ $CH_4$ production in the ocean mixed layer has received much attention, since $CH_4$ formation in the oxygenated ocean mixed layer challenge the paradigm that biological methanogenesis is a strictly anaerobic process." We further deleted the sentence "However, partitioning source categories to reduce uncertainties in the global CH4 budget is a major challenge (Saunois et al., 2016)."

Referee #1 Line 98: (Were cultures axenic? How was this determined? Sterile technique is not enough if bacteria are intrinsic to algal cultures. Bob Guillard told me this when I was using his culture collection. I personally don't think that there are anaerobic bacteria producing methane in rapidly photosynthesizing cultures, but one should be accurate.

Authors: We can't consider our approach as fully axenic and the reviewer is right that it is extremely difficult to grow algal cultures without bacteria. However, the algal cultures were diluted many times, resulting in exponential algal growth while minimizing bacterial cell density. This is a common practice to keep non-axenic algae cultures largely free of bacteria and it was applied in many other physiological algal studies before, which used non-axenic cultures. Please see also answers regarding comments by reviewer 2 (manuscript line 98ff and line 381), where we discuss a potential contribution of heterotrophs and/or methanogenic archaea. Briefly, the correlations we describe clearly show that CH4 production depends on algal growth. It is therefore highly unlikely that bacteria are solely responsible for CH4 production in our cultures. However, bacteria might be involved in the CH4 production process. One scenario which we cannot rule out would be a production of CH4 precursors by algae and a usage of these precursors by bacteria to produce CH4. While we think that this is less likely than CH4 production by algae alone, it would, if true, show that bacteria need algae-produced precursors to produce CH4. The latter scenario would be relevant in the field because algae co-exist with bacteria in the oceans. We have modified the Discussion and Abstract to make this clear. For more details see reply to reviewer #2 (manuscript line 98ff and line 381).

Referee #1 Line 115: When calculating the amount of methane produced, was fraction dissolved included? With a large headspace, this may be small but should be mentioned. Were samples equilibrated with headspace before methane measured? The

authors mention that oxygen was sometimes supersaturated, but was this relative to headspace or equilibration with ambient air?

Authors: The amount of dissolved $CH_4$ was not included. As requested we have calculated the dissolved $CH_4$ concentration by using the equation of Wiesenburg and Guinasso (1979). The dissolved fraction of $CH_4$ has now been included in our calculations and added to the total amount of $CH_4$ produced. As correctly stated by the referee the addition of the dissolved $CH_4$ fraction has only a marginal effect on the overall $CH_4$ production. Calculation of dissolved $CH_4$ is mentioned in the method section (2.6) and a new reference for calculating dissolved $CH_4$ was added (Wiesenburg and Guinasso, 1979) to the revised manuscript. Cultures were turned 30 seconds overhead prior to analysis to ensure equilibration between dissolved and headspace $CH_4$. In the preliminary equilibration experiments, we found that further shaking does not affect the $CH_4$ measurement and therefore all samples can be considered as equilibrated. We modified the sentence line 260: "The measured oxygen concentrations were always saturated or supersaturated relative to equilibration with ambient air (S.2)."

Referee #1 Line 133: Concentrations (final) of added substrates should be given for comparison with natural concentrations. If possible give concentrations of these substrates in medium at start of incubation with and without addition of substrate.

Authors: The final concentration of 13C-hydrogen carbonate (NaH13CO3) was 48.7 $\mu$mol L-1 and 10 $\mu$M for 13C2-DMS, 13C2-DMSO and 13C-MSO. Concentrations (final) of added substrates are given in the manuscript in line 133 for NaH13CO3 and at line 173 for 13C2-DMS, 13C2-DMSO and 13C-MSO. Cultures were grown in sterile filtered (0.2 $\mu$m Ø pore size) natural North Sea seawater (sampled off Helgoland, Germany) enriched in nutrients according to F/2 medium. The dissolved inorganic carbon (DIC) was 2152 $\pm$ 6 $\mu$mol L-1 (line 104). This value falls within the range of typical DIC concentrations of North Sea seawater. The added amount of NaH13CO3 corresponds to 2% of the DIC of the North Sea seawater. This information was added to the revised manuscript: "The DIC value falls within the range of typical DIC concentrations of North

Sea seawater. "We added two sentence to the section were we explain labeling experiments: "For stable carbon isotope experiments 48,7 $\mu$mol L-1 13C-hydrogen carbonate (NaH13CO3) in final concentration was added to the F/2 medium. The added amount of NaH13CO3 corresponds to 2% of the DIC of the North Sea seawater (2152 $\pm$ 6 $\mu$mol L-1), resulting in a theoretical calculated 13C value of DIC of +2014 $\pm$ 331‰."

Unfortunately the natural DMS, DMSO and MSO concentrations in our seawater were not determined. However, the global DMS mean concentration has been reported to be ca. 2 nM (Galí et al., 2018). A rough estimation can also be made for DMSO concentrations in the ocean as DMSO is generally present in concentrations 1–2 orders of magnitude greater than DMS (Lee et al., 1999). These estimates are also in line with data reported from a cruise of the western Pacific Ocean that were reported by Zindler et al. (2013). The average (total) DMS ,DMSP and DMSO concentrations were ca. 1 nM, 4 nM, and 16 nM for DMS, DMSP and DMSO respectively. Thus we conclude that the initial substrate concentration in the seawater is insignificant in comparison to the added amount (10$\mu$M), the latter being roughly two orders of magnitude higher than typically reported for oceanic concentrations (please see also reply to referee#2: line 172). Moreover, intracellular concentrations of methyl-sulfur compounds also play a significant role. We will discuss this issue below (see answer to next comment).

Referee #1 Line 327: If the labelled methyl groups yield only a small percentage (less than 1%) of total methane produced where is the other methane coming from? Is this result consistent with field observations that show only a weak link if any between DMS or DMSO and excess methane in surface water? This point needs more elaboration since the question of the source of excess methane in seawater has been plagued by studies that show methane can be produced by a process but that rates are far lower than are needed to explain natural surface water values. Here is where a link to ambient DMS, DMSO or MSO concentrations should be made. I think this point is a key issue.

Authors: Please note that the main reason for the isotope experiments was to unambiguously show that the tested compounds might be able to form CH4 under oxic conditions. The 13C-labeling experiment showed that DMS, DMSO, and MSO are potentially important methyl-precursors for CH4 but the contribution of these compounds to the overall CH4 production in cultures of E. huxleyi could not be determined in our experiments due to the complexity of the formation of these compounds in the algal cells. Hence, the stable isotope labeling approach should be considered as a proof of concept, showing that methyl groups of all tested substance serve as precursor compounds of CH4. Althoff et al. (2014) and Benzing et al. (2017) suggested a chemical reaction of DMSO, DMS and MSO that leads to CH4 formation in eukaryotes, especially, in marine algae containing elevated concentration of these compounds. We have therefore tested whether the methyl groups of these substances can actually be converted to CH4 in marine algae cultures. We made this point clearer in the discussion of the revised manuscript. The paragraph reads now: "The 13C-labeling experiment showed that DMS, DMSO, and MSO are potentially important methyl-precursors for CH4 but the contribution of these compounds to the overall CH4 production in cultures of E. huxleyi could not be determined in our experiments due to the complexity of the formation of these compounds in the algal cells. This can be illustrated by the following. The contribution of a substance to the total CH4 released is the product of both the added 13C-labeled fraction (added to the waters sample and uptake by the cells) and the internally formed fraction of these compounds (DMS, DMSO, and MSO) which will roughly show natural 13C abundance. Therefore the stable isotope value of CH4 will be diluted by the fraction of naturally formed methyl sulfur compounds in the algal cells and thus the contribution of DMS, DMSO, and MSO to CH4 formation can therefore not be estimated on the basis of their added amount alone. The 13CH4 quantity from conversion of added 13C labelled substance contributed 0.03% (13C2-DMSO) up to 0.84% (13C-MSO) to overall released CH4. However, even if the added 13C labelled compounds might only explain ≤ 1% of CH4 formed by the algae their overall contribution (including non-labelled sulfur compounds which we are not able to measure) might provide a much larger fraction of the released CH4. The intracellular DMS concentration can reach 1 mM (Sunda et al., 2002) in cells of E. huxleyi, while the concentration of added 13C2 -DMS was 0.01 mM in medium (final concentration). If intracellular 13C2 -DMS was in equilibrium with bulk seawater 13C2 -DMS and all CH4 would be produced from intracellular DMS, then the contribution of the 13C labeled compound would be about 1%. However, even if the biggest fraction of CH4 in algae cultures was not released by the 13C labelled substances, the significant increase in delta notation in 13C2-DMS, 13C2-DMSO and 13C-MSO treated cultures above the $\delta$13CH4 values of the control groups demonstrate that 13C labelled precursor substances were converted to CH4 by algal cultures (Fig.4 a-c). This is also indicated, when the absolute conversion quantities of 13C-labelled substance in algal cultures are considered: these were ca. nine (13C2-DMS), three (13C2-DMSO) and thirty (13C-MSO) times higher than in seawater control groups. Hence, the stable isotope labeling approach should be considered as a proof of concept, that methyl groups of all tested substance serve as precursor compounds of CH4."

We furthermore deleted the paragraph (line 341-354), since the main points regarding the CH4 conversion rates of 13C labeled compounds were discussed in the section above.

Referee #1 Line 400: Weller et al may have found a correlation between chlorophyll a and methane concentrations but there were many studies in the older literature (1970s and 80s) where no such correlation was observed. I recommend authors go back and read over some of these earlier papers and confirm that measured production rates from this study can support other previously observed methane fluxes. Also see thesis by Scranton (1977) where methane production was examined in cultures by several species including Emiliani huxleyi (called Coccolithus huxleyi in my thesis) and T. pseudonana. I observed methane production in a much less sophisticated experimental setup and concluded that natural populations of the algae I studied might be adequate to support the widespread supersaturations of methane seen in the open ocean (including in places where no dense algal blooms were observed). Perhaps your results

can be compared to mine or to other studies that report cell abundances and air-sea fluxes. A citation to a downloadable copy of my thesis is below. Scranton MI (1977) The marine geochemistry of methane. Citable URI https://hdl.handle.net/1912/1616. DOI10.1575/1912/1616.

Authors: We followed the recommendation of the reviewer (Mary Scranton) and compared the CH4 production rates of E. huxleyi reported by Scranton (1977) with those of our study. In line 392 we added: "We also compared the cellular CH4 production rates of E. huxleyi reported by Scranton (1977) with those of our study. Scranton (1977) reported a production rate of $2 \times 10$-10 nmol CH4 cell-1 hr-1. This value is close to the production rate of $1.4 \times 10$-10 nmol CH4 cell-1 hr-1 in our study. Scranton (1977) concluded from observed CH4 production rates in laboratory experiments that natural populations might be adequate to support the widespread supersaturations of CH4 seen in the open ocean." As the referee stated correctly the distribution of chlorophyll has not shown a consistent correlation with CH4 distributions in field studies.

The following section was added in Chapter 4.3: "In general, the distribution of chlorophyll has not shown a consistent correlation with CH4 distributions in field studies. There are studies were no correlation was observed (e.g. Lamontagne et al., 1975; Foster et al., 2006; Watanabe et al., 1995) or at least a correlation was found within a few depth profiles (Burke et al., 1983; Brooks et al., 1981). Many field measurements in oxygenated surface waters in marine and limnic environments have shown examples where elevated CH4 concentrations were spatially related to phytoplankton occurrence (e.g. Conrad and Seifer, 1988; Owens et al., 1991; Oudot et al., 2002; Damm et al., 2008; Grossart, et al., 2011; Weller et al., 2013; Zindler et al., 2013; Tang et al., 2014; Bogard et al., 2014; Rakowski et al., 2015). Taken together these studies suggest that phytoplankton is not the sole source of CH4 in oxygenated surface waters, but importantly they also suggest that phytoplankton is one of the sources of CH4. We therefore compared the CH4 production rates of our cultures with two field studies for the Pacific Ocean (Weller et al., 2013) and the Baltic Sea (Schmale et al., 2018) to evaluate the

potential relevance of algal CH4 production."

We followed the reviewer suggestion and added an additional comparison of our CH4 production rates by using field study data of Schmale et al., (2018). The respective section reads: "Schmale et al., (2018) reported CH4 enrichments that were observed during summer in the upper water column of the Gotland Basin, central Baltic Sea. They furthermore found that zooplankton is one but not the only CH4 source in the oxygenated upper waters. While the authors ruled out a major contribution of algae to the observed sub-thermocline CH4 enrichment because of the low sub-thermocline phytoplankton biomass, they considered a primary production associated CH4 formation as one likely source in the phytoplankton-rich mixed layer. The average phytoplankton carbon biomass of the mixed layer was approximately 600 $\mu$g L-1 (averaged from Fig. 9 in Schmale et al., 2018). For the reported average net CH4 production rate in the mixed layer (95 pmol CH4 L-1 d-1), we calculated that a production rate of 2.5 $\mu$g g-1 POC d-1 is required if the CH4 is produced by the algal biomass. This rate would be within the range of CH4 production rates observed in our study. These calculations should be considered as a first rough estimate to assess whether CH4 production rates of laboratory grown cultures can significantly contribute to CH4 supersaturation associated with phytoplankton. We did not distinguish between species and did not take into account environmental factors or the complexity of microbiological communities."

Minor issues

Referee #1: Equation 7: There should be a factor of 1000 to convert ratios to per mille values

Authors: We would like to keep equation 7 as is as it follows the recommendations by Coplen (2011) ("Guidelines and recommended terms for expression of stable isotope-ratio and gas-ratio measurement").

Referee #1 Figure 1: Plot control values here too.

Authors: Control groups in Figure 1 were plotted in the revised manuscript. In order to add control groups the unit was changed in concentration (ng CH4 L-1).

Referee #1 Line 268: Should it be "were applied" not "where applied"?

Authors: Yes. Corrected.

Referee #1 Line 308: Inoculation OF cells?

Authors: Yes. "of" was added.

References

[revised manuscript text omitted]

---

## Author Comment (AC3) · 14 Sep 2019

Point-by-point response to the issues raised by referee #2 (Anonymous)

We thank the reviewer for efforts in reviewing our manuscript and for the helpful comments which have improved the manuscript.

Referee #2 The present paper presents an interesting study about methane production under oxic conditions in marine environments. This so called "methane paradox" is a very important research field to understand methane emissions from oceans (and

lakes) and has recently received strong interest by a number of investigators from different scientific disciplines. The author presents data from incubation experiments conducted with three different algal species. Methane production rates were determined with different methanogenic substrates (13C-labled) using a stable isotope approach. A similar kind of studies was previously conducted for Emiliania huxleyi by Lenhart et al. (2016) and the isotope approach was successfully used in diverse investigations by Frank Keppler before to examine terrestrial methane production. The novel outcome in the present study is (1) that also other widespread haptophytes have the potential to produce methane under anoxic conditions; and (2) methylated sulphur compounds (e.g. DMS), that are known to be enriched in the investigated algae species, present potential substrates. In addition, the authors present an attempt to transfer their results to an algal bloom in the Pacific Ocean to discuss the potential relevance of algal methane production. The experiments are well thought out and the results present an additional piece in the complex puzzle. There are lots of little corrections needed and from my point of view some sentences need another structure to make the content more accessible for readers that are not familiar with the topic in detail (especially in the method section, e.g. PP, exponential growth rate). I will give a few examples below. Some minor and major points need to be addressed and I therefor recommend a publication after major revision.

Authors: We thank the referee for the positive evaluation of our manuscript and for the helpful comments. Requested changes were taken into account, as detailed in the following.

Referee #2: Line 95ff. The experimental design is very complex. A flowchart for the method section would be helpful for the reader.

Authors: We added a graphic/flowchart to the method section of the revised manuscript.

Referee #2: Line 98ff. How clean are the algal culture samples (purity)? Small differences in the degrees of contamination with archaea/bacteria (nitrogen limited bacteria, Line 69,Damm) between the cultures may have an impact on CH4 production rate. Does the web link give information about the purity of the culture?

Authors: Unfortunately, the weblink gives no detailed information about the purity of culture. We cannot consider our approach as axenic because it is extremely difficult to grow algal cultures without any bacteria. However, the algal cultures were diluted regularly, resulting in exponential algal growth and minimal bacterial growth. This is a common practice to keep non-axenic algae cultures largely free of bacteria (please see also answer to comment of reviewer #1 concerning manuscript line 98). We now mention that this cultures were non- axenic. We added the following sentence: "In order to keep non-axenic algae cultures largely free of bacteria, the cultures were diluted regularly, resulting in quasi constant exponential algal growth while minimizing bacterial cell density." However, we are aware that bacteria might play a role in CH4 production, but even if they did they still would depend on algal growth in our cultures as demonstrated by the following points.

1) The CH4 production rates decreased with decreasing algal growth rates: In batch cultures, the algae cultures undergo various stages of growth (see section 3.1, Fig. 1 a-c). Bacterial density increases tremendously when algae culture reach stationary growth phase and excretion of organic products from senescent alga cells together with the decomposition of cells is providing substratum for heterotrophs. This was described in literature (Salvesen et al., 2000) and is in line with our own experience with growing alga cultures in batch mode. In section 3.1 cultures have undergone transitionary growth phase leading up to the stationary phase. We calculated daily incremental CH4 production rates (not shown in the manuscript). The CH4 production rates of each species decreased with decreasing growth rates and decreased drastically when approaching stationary phase. This observation is the opposite of what we would have expected, if CH4 were mainly produced by bacteria. It would however be compatible with the idea that algae produce precursors which are subsequently used by bacteria

to produce CH4.

2) Light is a prerequisite for CH4 formation in algae cultures: Cultures of E. huxleyi, and P. globose were incubated under a day-night-cycle and continuous darkness. Methane concentrations did not increase when cultures were incubated in darkness while concentrations increased in cultures growing under a day-night-cycle. This is a strong indication that CH4 formation is dependent on the light-dependent metabolism of the algae, since the metabolism of heterotrophs or archaea is independent of light. While the latter conclusion does not rule out the "algae precursor scenario", our experimental setup makes it rather unlikely. In these experiments we inoculated high cell densities ($\approx$ 105 cells mL-1) because they were designed to be short term which requires a high start cell density to yield measurable production. Therefore the start conditions will have included a seawater replete with precursors. It is unlikely that the relatively few bacteria present should have become precursor-limited over a single dark phase. It is rather more likely that the pool of precursors was sufficient to sustain bacterial CH4 production over the dark phase. In this scenario an extra precursor production by cultures exposed to light would have been without effect on CH4 production.

3) It is highly unlikely that methanogenic archaea are the source of CH4 in cultures where CH4 is produced alongside oxygen (incubation under day-night-cycle). If archaea were the CH4 source we would have expected a higher CH4 production in the dark.

4) Selectively inhibition of algal growth reduced CH4 production rates: We compared emission rates of E. huxleyi that have been treated with and without 3-(3,4 dichlorophenyl)-1,1-dimethyl-urea (DCMU). DCMU acts as an inhibitor of photosynthesis (Wessels and Van Der Veen, 1956). Selectively inhibition of algal photosynthesis reduced both algal growth rate and CH4 production rates. In the inhibition experiments, the growth rate was only 29% of the uninhibited culture and the CH4 production rate dropped to 18% of the uninhibited culture. Since the inhibition effect of DCMU is very selective for algae (Francoeur et al., 2007) the result may indicate direct

CH4 production from algae. Although we regard it as unlikely, we cannot strictly rule out the "precursor-scenario": Bacteria use algae-derived precursors to produce CH4, and these bacteria require constant production of these precursors by algae. In other words, the precursor-production by algae is the rate limiting step of CH4 production by bacteria (as evident from points 1, 2, and 4 above). If true the CH4 production observed in our experiments would be the result of a "collaborative effort" which needs both partners, algae and bacteria. This would be a significant finding and prompt further research. Questions to be addressed would include: what are the precursors? Which algae can produce the precursors? Which bacteria can produce CH4 using these precursors? Is it possible to grow the respective algae without the bacteria (not all algal cultures can survive in an axenic state). Can the same CH4 production be achieved by growing the bacteria without algae and adding the precursors? This selection of questions would suffice for an entire research project. Meanwhile we are content with describing CH4 production that depends on algae, whether solely or in cooperation with bacteria. To sum up, our main finding is that CH4 production in mixed algae/bacteria cultures depends on algal growth and is not supported when algae become senescent. Future research will clarify whether algae alone produce CH4 or whether they produce precursors which in turn are used by bacteria to produce CH4. We have made this important point clear in the revised manuscript. We added this information in supplementary material. We now discuss the possible contribution of bacteria and archaea in the main text and refer to further discussion in the supplementary material (see discussion above). The revised the paragraph (Chapter 4.1) now reads:" The algal metabolites DMSP, DMS and DMSO are ubiquitous in marine surface layers and nanomolar concentrations were found in blooms of Chrysochromulina sp., P. globosa and E. huxleyi. Several field studies showed that these compounds are linked to CH4 formation in seawater (Damm et al., 2008; Zindler et al., 2013; Florez-Leiva et al., 2013). The authors proposed that DMSP and their degradation products DMSO and DMS are used by methylotrophic methanogenic archaea, inhabiting anoxic microsites, as substrates for methanogenesis. In addition it was reported that, if nitrogen is limited

but phosphorus is replete, marine bacteria might also use DMSP as a carbon source, thereby releasing CH4 (Damm et al., 2010). One scenario which we cannot rule out would be a production of CH4 precursors by algae and a usage of these precursors by bacteria to produce CH4. While we think that this is less likely than CH4 production by algae alone, it would, if true, show that bacteria need algae-produced precursors to produce CH4. The latter scenario would be relevant in the field because algae co-exist with bacteria in the oceans. Therefore bacteria might be involved in the CH4 production process in our cultures, but even if they were they still would depend on algal growth. For further discussion of a potential contribution of heterotrophs and/or methanogenic archaea see supplementary material. The correlations we describe in the supplementary material clearly show that CH4 production depends on algal growth. It is therefore highly unlikely that bacteria are solely responsible for CH4 production in our cultures."

Referee #2: Line 133. What is the difference in concentration of NaHCO3 between natural and inoculated water sample? Why did the authors added this amount of tracer? Should be mentioned.

Authors: Natural North Sea surface seawater contains ca 2000 $\mu$mol L-1 bicarbonate. We added 48,7 $\mu$mol L-1 13C-bicabonate , i.e. about 2 % of the natural concentration. This bicarbonate concentration was chosen for two reasons, one analytical, the other physiological. The physiological reason is that phytoplankton is sensitive to changes in seawater carbonate chemistry (see reviews on "ocean acidification"). We aimed at a negligible physiological effect of the added bicarbonate. The chosen bicarbonate concentration fulfills this criterion. The analytical reason is this: On the basis of the amount of added 13C-bicarbonate we calculated the theoretical $\delta$13C -DIC value (see also manuscript line 134). Based on the theoretical $\delta$13C -DIC value and from the previously determined CH4 increases in the cultures, the $\delta$13CH4 values can be estimated. The amount of 13C- bicarbonate was chosen on the basis of expected changes of $\delta$13CH4 values which were measured using GC-IRMS. A change of tenth to few hundred per mil is ideal regarding statistical issues (applying keeling plots for

source identification) but also concerning linearity issues of the isotope ratio mass spectrometer.

Referee #2: Line 141. Here, you should explain in more detail why an exponential growth rate is important to best compare CH4 formation between the experiments. From this sentence one could assume that Langer performed already methane production rate experiments with algae that indicated that exponential growth rates are important. From my point of view the activity of the cell is important for the turnover of these substrates and not their reproduction. It should be mentioned in the method or result section that microbial methane turnover takes place in the incubations and the production rates presented are minimum rates > because methane oxidation is not considered in the calculations (e.g. see methods in de Angelis and Lee). Authors: We agree with the reviewer that from a physiological point of view the activity of the cell is the relevant parameter here. But as detailed below our point is purely methodological, not physiological. We have clarified this in the revised manuscript: "Exponential growth is a prerequisite for calculating production on the basis of growth rate and quota (here CH4 quota). The point is a general, technical one, and is not confined to CH4 production. The papers by Langer et al. (2012, 2013) discuss this point in the context of batch culture experiments. Briefly, production on this account is the product of growth rate and quota (e.g. CH4, calcite, organic carbon). Production here is an integrated value, typically over many cell divisions. For this calculation of production to be meaningful a constant growth rate is required. The exponential growth phase fulfills this criterion whereas the transition phase and the stationary phase do not. Therefore production cannot be calculated meaningfully in the non-exponential phases. The problem can, however, be minimized by using small increments (one day) because growth rate can be regarded as quasi-constant (see also Lenhart et al., 2016). "

We agree with the reviewer that our calculated rates should be regarded as minimum rates because of microbial methane turnover. Please see also answer to comment on manuscript line 98ff.

Referee #2: Line 137. Can you explain if aggregates or sediment was visible in the incubation?

Authors: Cells were counted under bright field microscopy and we did not observe any aggregates.

Referee #2: Line 172. Why was exactly this amount of substrate (DMS...) injected and is this comparable with natural environments (concentrations). Substrate concentrations definitely affect the turnover and the addition of tracers/substrates should not impact the sample too drastically. Why did the authors did not applied MET (and DMSP) as a precursor that was tested before successfully by Lenhart et al.?

Authors: The amount added was chosen based on the practical experience from previous experiments with E. huxleyi and methionine (Lenhart et al., 2016), so that the expected $\delta$13CH4 fall within a measurable range with statistical significance. The growth of algae was not effected and changes of the overall CH4 production did not change by the addition of substrate (within error of measurement). The added amount (10$\mu$M) of methylated sulfur compounds (13C2-DMS, 13C2-DMSO or 13C-MSO) was higher than those expected in ocean water samples (please see also answer to reviewer #1 comment on manuscript line 133). However, the intracellular concentrations of these compounds can reach mM levels (Keller, 1989; Rafel et al.,1998; Keller et al., 1999; Sunda et al., 2002), which is two orders of magnitude higher than the added concentration of 10 $\mu$M (final) in our experiments. For this reason, it can be expected that the amount of 13C labelled substance taken up by the algal cells is low in relation to the amount of methylated sulfur compounds what they synthesize during metabolism. The turnover of DMS, DMSO or MSO (including non-labeled compounds) to CH4 could not be determined on the basis of their added amount of 13C-labelled substance in cultures of E. huxleyi. Please see answers regarding comments of reviewer#1 (manuscript 327), where we discuss this issue in detail. Neither was this the goal of the experiment. However, it can be determined exactly how the ratio of 13C in CH4 increases, when 13C labeled methylated sulfur compounds were added (Fig. 4 a-c). It has therefore

been shown that methyl groups of these compounds can be converted into CH4 in algal cultures. Due to time constraints, we omitted the methionine treatment. Unfortunately, isotopically labelled 13C2 DMSP was not commercially available. Moreover, this compound could not be synthesized in our laboratories.

Referee #2: Line 327ff. Is it possible that a natural microbial community is needed for the turnover of these substances to methane? If the incubations are without contaminations (sterile filtered seawater, pure culture), the production rates might be low because of the missing community. The algae may only provide the precursors. Might be a point that could be discussed here.

Authors: Please see reply to referee #2: line 98ff .

Referee #2: Line 381. Argumentation is difficult. Only because Lenhart could prove a contamination-free incubations, this result cannot be transferred to all the incubation that will be performed by the working group afterwards. See comments/concerns to this topic above. Since the argumentation is difficult to follow, I suggest to discuss this topic less dominant and integrate this part somewhere else (not under a separate title). Also 50% of the text is nearly copied from the introduction (doubling!).

Authors: Agreed. This section has been modified according the referee's suggestions. Please see also reply to referee #2: line 98ff.

Referee #2: Line 335. I have a different impression. Figure 4a: At day 2 the d13C values are very close to each other. In Figure 4b all values from beginning to the end of the incubation time are very similar. Only Figure 4c shows a clear difference between culture and control over the course of the experiment. Add in the figure caption that also controls are plotted, not only results from cultures.

Authors: $\delta$13CH4 values for 13C2-DMS are presented in Figure 4 a. We found that DMS is also converted chemically in sterile filtered seawater. This is in line with observations of Zhang et al. (2015) (please see manuscript line 332-339). However, the

formation rates are very low and only become obvious when applying sensitive stable isotope labelling techniques. We agree with the reviewer that the $\delta$13CH4 values of the DMS spiked seawater group and the DMS spiked algae group are very close to each other up to day 2. A section regarding control values was added to the revised manuscript (Chapter 4.1) :" The 13C2-DMS spiked seawater group and the 13C2-DMS spiked algae group are very close to each other up to day 2 (see Fig.3a and Fig.4a). For this time period, it can be assumed that the chemical conversion has taken place in both samples to the same extent, since the samples are relatively similar, because the algal cell density is only 5% (day 2) of the final cell density. However, the following days (day 3 to day 6), when algal cell density increased drastically, the $\delta$13CH4 values of the algae cultures also increased significantly compared with $\delta$13CH4 values of the seawater. This clearly indicates that conversion of 13C2-DMS to CH4 increases with increasing cell counts."

Referee #2: Chapter 4.3 I would recommend to perform an additional calculation to show that algal CH4 production is an important mechanism that can explain air/sea methane fluxes and methane enrichments. For example Schmale et al. (2018) gives detailed data about phytoplankton biomass (e.g. Prymnesiales) and production rates needed to maintain air/sea fluxes and subthermocline methane enrichments. There are probably also other papers available that could be used for such calculation.

Authors: We followed the recommendation and used the detailed data of Schmale et al. (2018) to estimate a possible contribution of algal CH4 production to the CH4 production rate in the field. An additional calculation was added to Chapter 4.3 in the revised manuscript. See reply to referee #1: line 400.

Minor issues:

Referee #2: Title: I recommend writing "potential relevance for the environment". A direct Transfer of laboratory studies/results into field observations is difficult.

Authors: Change applied.

Referee #2: Line 24. Please also give the productions rates per cell in the abstract. Temperature is not needed to be mentioned in the abstract.

Authors: Changes applied.

Referee #2: Line 27ff. It should be mentioned here that the conversion of methylated sulphur compounds to methane was only responsible for less than 1% of the observed methane production (line 327ff).

Authors: We do not think that this information is important for the reader. The information might misleading here. Please see answers regarding comments of reviewer#1 (manuscript line 327), where we discuss this issue in detail.

Referee #2: Line 26-29. The word "clearly" is used to often.

Authors: We have rephrased line 26-29.

Referee #2: Line 30. "Relevance for the environment" is one major issue in the title but is reduced here to a little sentence. This part should be extended.

Authors: We have emphasized this issue in the abstract as requested. We added the following sentence: "By comparing the algae $CH_4$ production rates with two field studies form the Pacific Ocean and the Baltic Sea we concluded that $CH_4$ production could likely contributing to $CH_4$ oversaturation in oxic waters."

Referee #2: Line 49. How can "emissions from freshwater" explain the $CH_4$ concentration in ocean surface water?

Authors: We deleted "emissions from freshwater and" from line 49.

Referee #2: Line 50. Shorten the sentence and delete "that has been often: : :". "Well-known" means "often reported"

Authors: We have appropriately modified the sentence.

Referee #2: Line 55-58. This paragraph should be moved to line 46. It might be better

to start with this overall review before listing the recent specific studies to oxic methane production in lakes and ocean.

Authors: As requested, we have restructured this section.

Referee #2: Line 60. May also mention Valle and Karl (2014) who used in situ MPn concentration in a 14C approach and showed that dissolved MPn in surface waters cannot account for methane oversaturation.

Authors: The revised manuscript contains the results of del Valle and Karl (2014) as well as the results of Repeta et al. (2016), who, on the other hand, reported that the cycling of the organic matter phosphonate inventory could be sufficient to support the total atmospheric CH4 flux at their study site.

Referee #2: Line 98. A bracket is missing (RC: : :). Is it clear for the reader for what the web link is good for?

Authors: The bracket was added. We have added the names of the culture collections to the front of the weblink to make it clear that the links lead to the respective collections.

Referee #2: Line 102. Delete "in" in front of "natural"

Authors: "in was" deleted.

Referee #2: Line 110ff. Why did the authors used different volumes (medium and headspace)?

Authors: For practical reasons. We have limited space in the climatic chamber and a limited supply of natural seawater, therefore the size of the vessels were adjusted.

Referee #2: Line 119. What is meant with "main cultures"? Is this the investigated culture in the incubation?

Authors: Yes, it means the culture that was studied during the incubation. We rephrased this sentence to make it clear.

Referee #2: Line 119ff. I would suggest to transfer the cell densities to the result section (3.1).

Authors: Changes applied as suggested.

Referee #2: Line 122. E. huxleyi was sampled daily! What do you mean with overall sampling interval: 9,11,6 days? Why this odd order? And why did you sample the cultures in different intervals? If there is a reason for that it should be explained.

Authors: Overall sampling interval means the incubation time (from inoculation to the end of the experiment/incubation) for each species, that correspond to the sampling time. The incubation time varies from species to species and depends on the growth rate and the cell density in the stationary phase. The stationary phase for each species is dependent on a species specific cell density. As the species grow at different rates the sampling intervals differ. E. huxleyi has by far the highest growth rate and was sampled daily. Chrysochromulina sp. and P. globosa grow slower and were sampled at longer intervals. We revised this section in the manuscript and provided explanations.

Referee #2: Line 128. Why only three data points for E. huxleyi. From Figure 1c it seems to be plausible to use four.

Authors: The phase of exponential growth (from which $\mu$ was calculated) was defined by the cell densities which correspond to the best fit of the exponential regression. The fourth data point clearly deviates from exponential regression.

Referee #2: Line 131. It is always worth to have a repetition to support the previous results.

Authors: Due to the time constraints, we decided not repeat the experiments with stable isotope measurements that were already done by Lenhart et al. (2016).

Referee #2: Line 134. The delta is missing in d13C

Authors: Corrected.

Referee #2: Line 135. Suggest to write: ": : :values of the methane precursor: : :"

Authors: The expression "source" is usually used in combination with Keeling plots. We would therefore like to keep the expression "source".

Referee #2: Line 144. ": : :measured at the end: : :"

Authors: Corrected.

Referee #2: Line 145. Suggest to write "For this additional experiment: : :"

Authors: Agreed. Changes applied.

Referee #2: Line 146. Suggest putting the cell densities in the result section (see above).

Authors: Changes applied as suggested.

Referee #2: Line 153. "ag" is the abbreviation for what?

Authors: Please see bracket in line 153. The unit "ag" means 10-18 g. The SI prefix "a" stand for atto (10-18).

Referee #2: Line 158. The program "Image J" is produced by which company?

Authors: "Image J" is an open source software. A reference was added.

Referee #2: Line 174. I still think that cell densities should be implemented in the result section (see above).

Authors: Changes applied as suggested.

Referee #2: Line 176. The target/design of the experiment in section 3.2 is still un-known!

Authors: We have revised section 3.2 to make the experimental target and design clearer. Please see also the answer to the comment concerning manuscript line 95ff.

Referee #2: Line 189. Analyzed. See line 180

Authors: Corrected.

Referee #2: Line 193. Write: ": : : (based on three: : :)".

Authors: We changed line 193 as requested.

Referee #2: Line 204. Delete: ": : :at a temperature: : :". Here and in some other parts of the result section you mention details that were mentioned before in the method section. I would start with a sentence that makes clear that you are talking about the incubation with 13C-labelled hydrogen carbon (2.3).

Authors: Line 204 "at a temperature" was deleted. The result section was revised to avoid repetition of details from the method section. An introductory sentence was added in section 2.3 as suggested.

Referee #2: Line 208. Also here delete the repeated information: "These rates were obtained: : :". Check the entire result section to avoid redundancy.

Authors: The result section was revised to avoid repetition of details from the method section.

Referee #2: Line 211. The cell density should only be mentioned here and not in the method section!

Authors: All final cell densities were removed from the method section.

Referee #2: Line 214. Where is the control group plotted? Figure 2. Black and blue dots are difficult to distinguish. Even if it is "only" the control sample – make the visibility easier. The x-axis should be 1/CH4. Right?

Authors: Control groups in Figure 1 were plotted in the revised manuscript. Please also look at the answer to referee#1's comment regarding figure 1. Figure 2 was revised: by changing colors and correcting labeling of x-axis. Yes, "1/CH4" is correct.

Referee #2: Line 249. See above. Not clear why the exponential growth phase is important and not the cell activity.

Authors: Please see the answer above to the comment concerning manuscript line 141.

Referee #2: Line 251. The equation is already described above (2 and 4). Avoid doubling. See comment above.

Authors: We revised this section in order to avoid doubling. The sentence "By doing so the CH4 production rate is the product of exponential growth rate $\mu$ and cellular or POC quota." was removed from line 251.

Referee #2: Line 254. The sentence should end with (Tab. 1).

Authors: Corrected.

Referee #2: Line 256. "community level" sounds odd in this context. May you can find a better description.

Authors: This term was used in the context of the production potential which was established by Gafar et al. (2018). The phrase "community level" is also used in this context by Gafa et al. (2018). We would therefore like to keep this phrase in order to avoid misunderstandings.

Referee #2: Line 266. It starts again with information that was mentioned before in the methods.

Authors: We removed this information from line 266.

Referee #2: Line 271. The sentence should end with "(Fig. 4)".Figure 4b. Change the x-axis to 13C2 (add 2).Figure 4c. Change the x-axis to MSO (not MES, see caption).

Authors: Line 271 and Figure 4c were corrected.

Referee #2: Line 279. Add the control sample in the Figure.

Authors: Control groups were added.

Referee #2: Line 302. In the present study only the turnover of 13C-hydrogen carbonate by two algal species was investigate. Lenhart applied the isotope technique for E. huxleyi.

Authors: We have corrected this sentence.

Referee #2: Line 307. (with highest cell numbers) is out of context. Please rephrase.

Authors: The phrase "(with highest cell numbers)" was replaced by "(where the POC content is highest)".

Referee #2: Line 333. In future investigations I would suggest a dark incubation to exclude methane production by UV or visible light (line 70ff).

Authors: This topic is currently being investigated by us. Please see also answer to the comment of referee #2 concerning manuscript line 98. There we discuss a dark incubation experiment. In addition we discuss there a experiment with and without inhibition of algae by DCMU.

Referee #2: Line 352ff. Did Althoff really proved that the "reactivity" is the driving force in her experiments? Or are point 1 (label concentration) and 2 (penetration) also possible explanations for her observations?

Authors: Althoff et al. (2014) used a defined chemical system to study the conversion of methylated sulfur compounds to CH4. It turned out that the yield was not the same for all substances under otherwise identical reaction conditions. The CH4 yield was therefore also dependent on the substance and thus its reaction behavior.

Referee #2: Line 360ff. Sentence too complex. Devide in two parts.

Authors: As requested, we have reformulated and restructured this sentence.

Referee #2: Line 363 and 365. Too often "furthermore". Rephrase.

Authors: Changes applied.

Referee #2: Line 391. Interesting. But it needs be explained in more detail why the growth rate impacts the methane production. See above.

Authors: A detailed explanation was added to the method section. Please see reply referee #2: line 141.

Referee #2: Line 411. Include/explain why PP is meaningful parameter.

Authors: Further explanation regarding the meaning of PP was added to the manuscript. Line 410 now reads: "Gafar et al. (2018) suggested the production potential (PP), as opposed to cellular production as a biogeochemically meaningful parameter because the PP includes the impact of growth rates on cell densities in an exponentially growing community whereas cellular production rates do not."  

References

Damm, E., Kiene, R. P., Schwarz, J., Falck, E., and Dieckmann, G.: Methane cycling in Arctic shelf water and its relationship with phytoplankton biomass and DMSP, Marine Chemistry, 109, 45-59, 10.1016/j.marchem.2007.12.003, 2008.

Damm, E., Helmke, E., Thoms, S., Schauer, U., Nöthig, E., Bakker, K., and Kiene, R. P.: Methane production in aerobic oligotrophic surface water in the central Arctic Ocean, Biogeosciences, 7, 1099–1108, doi:10.5194/bg-7-1099-2010, 2010.

del Valle, D. A., and Karl, D. M.: Aerobic production of methane from dissolved water-column methylphosphonate and sinking particles in the North Pacific Subtropical Gyre, Aquatic Microbial Ecology, 73, 93-105, 2014.

Florez-Leiva, L., Damm, E., and Farías, L.: Methane production induced by dimethyl-sulfide in surface water of an upwelling ecosystem, Progress in oceanography, 112, 38-48, 2013.

Francoeur, S. N., Johnson, A. C., Kuehn, K. A., and Neely, R. K.: Evaluation of the

efficacy of the photosystem II inhibitor DCMU in periphyton and its effects on nontarget microorganisms and extracellular enzymatic reactions, Journal of the North American Benthological Society, 26, 633-641, 10.1899/06-051.1, 2007.

Keller, M. D.: Dimethyl sulfide production and marine phytoplankton: the importance of species composition and cell size, Biological oceanography, 6, 375-382, 1989.

Keller, M., Kiene, R., Matrai, P., and Bellows, W.: Production of glycine betaine and dimethylsulfoniopropionate in marine phytoplankton. I. Batch cultures, Marine Biology, 135, 237-248, 1999.

Langer, G., Oetjen, K., and Brenneis, T.: Calcification of Calcidiscus leptoporus under nitrogen and phosphorus limitation, Journal of Experimental Marine Biology and Ecology, 413, 131-137, 2012.

Langer, G., Oetjen, K., and Brenneis, T.: Coccolithophores do not increase particulate carbon production under nutrient limitation: A case study using Emiliania huxleyi (PML B92/11), Journal of experimental marine biology and ecology, 443, 155-161, 2013.

Lenhart, K., Klintzsch, T., Langer, G., Nehrke, G., Bunge, M., Schnell, S., and Keppler, F.: Evidence for methane production by the marine algae Emiliania huxleyi, Biogeosciences, 13, 3163-3174, 10.5194/bg-13-3163-2016, 2016.

Rafel, S. Â., Angela, D. H., Gillian, M., and Peter, S. L.: Particulate dimethyl sulphoxide in seawater: production by microplankton, Marine Ecology Progress Series, 167, 291-296, 1998.

Repeta, D. J., Ferrón, S., Sosa, O. A., Johnson, C. G., Repeta, L. D., Acker, M., DeLong, E. F., and Karl, D. M.: Marine methane paradox explained by bacterial degradation of dissolved organic matter, Nature Geoscience, 9, 884, 10.1038/ngeo2837, https://www.nature.com/articles/ngeo2837#supplementary-information, 2016.

Sunda, W., Kieber, D. J., Kiene, R. P., and Huntsman, S.: An antioxidant function for DMSP and DMS in marine algae, Nature, 418, 317-320, 10.1038/nature00851, 2002.

Wessels, J. S. C., and van der Veen, R.: The action of some derivatives of phenylurethan and of 3-phenyl-1,1-dimethylurea on the Hill reaction, Biochimica et Biophysica Acta, 19, 548-549, ://doi.org/10.1016/0006-3002(56)90481-4, 1956.

Zhang, Y., and Xie, H.: Photomineralization and photomethanification of dissolved organic matter in Saguenay River surface water, Biogeosciences, 12, 6823-6836, 10.5194/bg-12-6823-2015, 2015.

Zindler, C., Bracher, A., Marandino, C. A., Taylor, B., Torrecilla, E., Kock, A., and Bange, H. W.: Sulphur compounds, methane, and phytoplankton: interactions along a north-south transit in the western Pacific Ocean, Biogeosciences, 10, 3297-3311, 10.5194/bg-10-3297-2013, 2013

---

## Author Response (AR1)

**Point-by-point response to the issues raised by referee#1 (Mary Scranton)**

We thank the referee for the constructive comments and suggestions which have helped to improve the manuscript.

***Referee #1*** *(referee's comments are in italics)*

*This paper presents an interesting discussion about the importance of methane production by several species of algae under aerobic conditions in the ocean. The authors' experiments are original and convincing but I think they overstate (or ignore) the extent to which this process can result in methane excess concentrations in open ocean surface water. In turn the minor role of that excess production to the atmospheric methane budget is not clearly explained. Below are some substantive criticisms and some minor corrections.*

Authors: We appreciate the positive evaluation of our manuscript. The criticisms are addressed and corrections are made below.

**Referee #1** *Line 17: The abstract indicates that the importance of oceanic methane production to the global methane budget is unknown but this is not discussed further in article and is misleading in any case since the ocean is known to be a very small*

*contributor to the atmosphere. I am tired of proposals and papers that use the atmospheric methane budget to justify all studies of basic methane geochemistry. Surely it is enough to note a widespread and unexplained phenomenon which one is trying to explain mechanistically. I suggest adding a sentence or two to the introduction indicating why you are bothering to do this study and de-emphasizing how it might affect global methane budget. You are better off being straightforward and admitting that the real question is that methane is known to be produced in the oxic oceanic mixed layer and after more than 40 years*

*no one really understands why. Give some idea of what actual flux of methane to atmosphere from ocean is thought to be. This HAS been calculated a number of times.*

Authors: We agree with the referee and thus have modified the Abstract and Introduction. The first sentence of the abstract now reads:

"Methane ($CH_4$) production within the oceanic mixed layer is a widespread phenomenon, but the underlying mechanisms are still under debate."

We further added two sentences to the introduction:

"The world's oceans are considered to be a minor source of CH4 to the atmosphere (1-3 %, Saunois et al., 2016). However, in recent years the widespread occurrence of in situ $CH_4$ production in the ocean mixed layer has received much attention, since $CH_4$ formation in the oxygenated ocean mixed layer challenge the paradigm that biological methanogenesis is a strictly anaerobic process."

We further deleted the sentence "However, partitioning source categories to reduce uncertainties in the global $CH_4$ budget is a major challenge (Saunois et al., 2016)."

**Referee #1** *Line 98: (Were cultures axenic? How was this determined? Sterile technique is not enough if bacteria are intrinsic to algal cultures. Bob Guillard told me this when I was using his culture collection. I personally don't think that there are anaerobic bacteria producing methane in rapidly photosynthesizing cultures, but one should be accurate.*

Authors: We can't consider our approach as fully axenic and the reviewer is right that it is extremely difficult to grow algal cultures without bacteria. However, the algal cultures were diluted many times, resulting in exponential algal growth while minimizing bacterial cell density. This is a common practice to keep non-axenic algae cultures largely free of bacteria and it was applied in many other physiological algal studies before, which used non-axenic cultures. Please see also answers regarding comments by reviewer 2 (manuscript line 98ff and line 381), where we discuss a potential contribution of heterotrophs and/or methanogenic archaea. Briefly, the correlations we describe clearly show that $CH_4$ production depends on algal growth. It is therefore highly unlikely that bacteria are solely responsible for $CH_4$ production in our cultures. However, bacteria might be involved in the $CH_4$ production process. One scenario which we cannot rule out would be a production of $CH_4$ precursors by algae and a usage of these precursors by bacteria to produce $CH_4$. While we think that this is less likely than $CH_4$ production by algae alone, it would, if true, show that bacteria need algae-produced precursors to produce $CH_4$. The latter scenario would be relevant in the field because algae co-exist with bacteria in the oceans. We have modified the Discussion and Abstract to make this clear. For more details see reply to reviewer #2 (manuscript line 98ff and line 381).

**Referee #1** *Line 115: When calculating the amount of methane produced, was fraction dissolved included? With a large headspace, this may be small but should be mentioned. Were samples equilibrated with headspace before methane measured? The authors mention that oxygen was sometimes supersaturated, but was this relative to headspace or equilibration with ambient air?*

Authors: The amount of dissolved $CH_4$ was not included. As requested we have calculated the dissolved $CH_4$ concentration by using the equation of Wiesenburg and Guinasso (1979). The dissolved fraction of $CH_4$ has now been included in our calculations and added to the total amount of $CH_4$ produced. As correctly stated by the referee the addition of the dissolved $CH_4$ fraction has only a marginal effect on the overall $CH_4$ production. Calculation of dissolved $CH_4$ is mentioned in the method section (2.6) and a new reference for calculating dissolved $CH_4$ was added (Wiesenburg and Guinasso, 1979) to the revised manuscript.

Cultures were turned 30 seconds overhead prior to analysis to ensure equilibration between dissolved and headspace $CH_4$. In the preliminary equilibration experiments, we found that further shaking does not affect the $CH_4$ measurement and therefore all samples can be considered as equilibrated.

We modified the sentence line 260: "The measured oxygen concentrations were always saturated or supersaturated relative to equilibration with ambient air (Fig. S2)."

**Referee #1** *Line 133: Concentrations (final) of added substrates should be given for comparison with natural concentrations. If possible give concentrations of these substrates in medium at start of incubation with and without addition of substrate.*

Authors: The final concentration of $^{13}$C-hydrogen carbonate (NaH$^{13}$CO$_3$) was 48.7 µmol L$^{-1}$ and 10 µM for $^{13}$C$_2$-DMS, $^{13}$C$_2$-DMSO and $^{13}$C-MSO. Concentrations (final) of added substrates are given in the manuscript in line 133 for NaH$^{13}$CO$_3$ and at line 173 for $^{13}$C$_2$-DMS, $^{13}$C$_2$-DMSO and $^{13}$C-MSO.

Cultures were grown in sterile filtered (0.2 µm Ø pore size) natural North Sea seawater (sampled off Helgoland, Germany) enriched in nutrients according to F/2 medium. The dissolved inorganic carbon (DIC) was 2152 ± 6 µmol L$^{-1}$ (line 104). This value falls within the range of typical DIC concentrations of North Sea seawater. The added amount of NaH$^{13}$CO$_3$ corresponds to 2% of the DIC of the North Sea seawater. This information was added to the revised manuscript: "The DIC value falls within the range of typical DIC concentrations of North Sea seawater. "We added two sentence to the section were we explain labeling experiments: "For stable carbon isotope experiments 48,7 µmol L$^{-1}$ $^{13}$C-hydrogen carbonate (NaH$^{13}$CO$_3$) in final concentration was added to the F/2 medium. The added amount of NaH$^{13}$CO$_3$ corresponds to 2% of the DIC of the North Sea seawater (2152 ± 6 µmol L$^{-1}$), resulting in a theoretically calculated $^{13}$C value of DIC of +2014 ± 331‰."

Unfortunately the natural DMS, DMSO and MSO concentrations in our seawater were not determined. However, the global DMS mean concentration has been reported to be ca. 2 nM (Galí et al., 2018). A rough estimation can also be made for DMSO concentrations in the ocean as DMSO is generally present in concentrations 1–2 orders of magnitude greater than DMS (Lee et al., 1999). These estimates are also in line with data reported from a cruise of the western Pacific Ocean that were reported by Zindler et al. (2013). The average (total) DMS ,DMSP and DMSO concentrations were ca. 1 nM, 4 nM, and 16 nM for

DMS, DMSP and DMSO respectively. Thus we conclude that the initial substrate concentration in the seawater is insignificant in comparison to the added amount (10µM), the latter being roughly two orders of magnitude higher than typically reported for oceanic concentrations (please see also reply to referee#2: line 172). Moreover, intracellular concentrations of methyl-sulfur compounds also play a significant role. We will discuss this issue below (see answer to next comment).

***Referee #1 Line 327:*** *If the labelled methyl groups yield only a small percentage (less than 1%) of total methane produced where is the other methane coming from? Is this result consistent with field observations that show only a weak link if any between DMS or DMSO and excess methane in surface water? This point needs more elaboration since the question of the source of excess methane in seawater has been plagued by studies that show methane can be produced by a process but that rates are far lower than are needed to explain natural surface water values. Here is where a link to ambient DMS, DMSO or*

*MSO concentrations should be made. I think this point is a key issue.*

Authors: Please note that the main reason for the isotope experiments was to unambiguously show that the tested compounds might be able to form CH$_4$ under oxic conditions. The $^{13}$C-labeling experiment showed that DMS, DMSO, and MSO are potentially important methyl-precursors for CH$_4$ but the contribution of these compounds to the overall CH$_4$ production in cultures of *E. huxleyi* could not be determined in our experiments due to the complexity of the formation of these compounds in the algal cells. Hence, the stable isotope labeling approach should be considered as a proof of concept, showing that methyl groups of all tested substance serve as precursor compounds of $CH_4$. Althoff et al. (2014) and Benzing et al. (2017) suggested a chemical reaction of DMSO, DMS and MSO that leads to $CH_4$ formation in eukaryotes, especially, in marine algae containing elevated concentration of these compounds. We have therefore tested whether the methyl groups of these substances can actually be converted to $CH_4$ in marine algae cultures. We made this point clearer in the discussion of the revised manuscript. The paragraph reads now:

"The [13]C-labelling experiment showed that DMS, DMSO, and MSO are potentially important methyl-precursors for $CH_4$ but the contribution of these compounds to the overall $CH_4$ production in cultures of *E. huxleyi* could not be determined in our experiments due to the complexity of the formation of these compounds in the algal cells. This can be illustrated by the following. The contribution of a substance to the total $CH_4$ released is the product of both the added [13]C-labeled fraction (added to the waters sample and uptake by the cells) and the internally formed fraction of these compounds (DMS, DMSO, and MSO) which will roughly show natural [13]C abundance. Therefore the stable isotope value of $CH_4$ will be diluted by the fraction of naturally formed methyl sulfur compounds in the algal cells and thus the contribution of DMS, DMSO, and MSO to $CH_4$ formation can therefore not be estimated on the basis of their added amount alone. The $^{13}CH_4$ quantity from conversion of added [13]C labelled substance contributed 0.03% ($^{13}C_2$-DMSO) up to 0.84% ($^{13}C$-MSO) to overall released $CH_4$. However, even if the added [13]C labelled compounds might only explain $\leq 1\%$ of $CH_4$ formed by the algae their overall contribution (including non-labelled sulfur compounds which we are not able to measure) might provide a much larger fraction of the released $CH_4$. The intracellular DMS concentration can reach 1 mM (Sunda et al., 2002) in cells of *E. huxleyi*, while the concentration of added $^{13}C_2$ -DMS was 0.01 mM in medium (final concentration). If intracellular $^{13}C_2$ -DMS was in equilibrium with bulk seawater $^{13}C_2$ -DMS and all $CH_4$ would be produced from intracellular DMS, then the contribution of the [13]C labeled compound would be about 1%. However, even if the biggest fraction of $CH_4$ in algae cultures was not released by the [13]C labeled substances, the significant increase in delta notation in $^{13}C_2$-DMS, $^{13}C_2$-DMSO and $^{13}C$-MSO treated cultures above the $\delta^{13}CH_4$ values of the control groups demonstrate that [13]C labelled precursor substances were converted to $CH_4$ by algal cultures (Fig.4 a-c).

This is also indicated, when the absolute conversion quantities of [13]C-labelled substance in algal cultures are considered: these were ca. nine ($^{13}C_2$-DMS), three ($^{13}C_2$-DMSO) and thirty ($^{13}C$-MSO) times higher than in seawater control groups. Hence, the stable isotope labeling approach should be considered as a proof of concept, that methyl groups of all tested substance serve as precursor compounds of $CH_4$."

We furthermore deleted the paragraph (line 341-354), since the main points regarding the $CH_4$ conversion rates of [13]C labeled compounds were discussed in the section above.

*Referee #1 Line 400:* *Weller et al may have found a correlation between chlorophyll a and methane concentrations but there were many studies in the older literature (1970s and 80s) where no such correlation was observed. I recommend authors go back and read over some of these earlier papers and confirm that measured production rates from this study can support other*

*previously observed methane fluxes. Also see thesis by Scranton (1977) where methane production was examined in cultures by several species including Emiliani huxleyi (called Coccolithus huxleyi in my thesis) and T. pseudonana. I observed methane production in a much less sophisticated experimental setup and concluded that natural populations of the algae I studied might be adequate to support the widespread supersaturations of methane seen in the open ocean (including in places where no dense algal blooms were observed). Perhaps your results can be compared to mine or to other studies that report cell*

*abundances and air-sea fluxes. A citation to a downloadable copy of my thesis is below. Scranton MI (1977) The marine geochemistry of methane. Citable URI https://hdl.handle.net/1912/1616. DOI10.1575/1912/1616.*

Authors: We followed the recommendation of the reviewer (Mary Scranton) and compared the $CH_4$ production rates of *E. huxleyi* reported by Scranton (1977) with those of our study.

In line 392 we added: "We also compared the cellular $CH_4$ production rates of *E. huxleyi* reported by Scranton (1977) with those of our study. Scranton (1977) reported a production rate of $2\times10^{-10}$ nmol $CH_4$ cell$^{-1}$ hr$^{-1}$. This value is close to the production rate of $1.6 \times 10^{-10}$ nmol $CH_4$ cell$^{-1}$ hr$^{-1}$ in our study. Scranton (1977) concluded from observed $CH_4$ production rates in laboratory experiments that natural populations might be adequate to support the widespread supersaturations of $CH_4$ seen in the open ocean."

As the referee stated correctly the distribution of chlorophyll has not shown a consistent correlation with $CH_4$ distributions in field studies.

The following section was added in Chapter 4.3: "In general, the distribution of chlorophyll has not shown a consistent correlation with $CH_4$ distributions in field studies. There are studies in which no correlation was observed (e.g. Lamontagne et al., 1975; Foster et al., 2006; Watanabe et al., 1995) or a correlation was found within a few depth profiles (Burke et al., 1983;

Brooks et al., 1981). Many field measurements in oxygenated surface waters in marine and limnic environments have shown examples of elevated $CH_4$ concentrations spatially related to phytoplankton occurrence (e.g. Conrad and Seifer, 1988; Owens et al., 1991; Oudot et al., 2002; Damm et al., 2008; Grossart, et al., 2011; Weller et al., 2013; Zindler et al., 2013; Tang et al., 2014; Bogard et al., 2014; Rakowski et al., 2015). Taken together these studies suggest that phytoplankton is not the sole source of $CH_4$ in oxygenated surface waters, but importantly they also suggest that phytoplankton is one of the sources of $CH_4$.

We therefore compared the $CH_4$ production rates of our cultures with two field studies for the Pacific Ocean (Weller et al., 2013) and the Baltic Sea (Schmale et al., 2018) to evaluate the potential relevance of algal $CH_4$ production. "

We followed the reviewer suggestion and added an additional comparison of our $CH_4$ production rates by using field study data of Schmale et al., (2018).

The respective section reads: "Schmale et al., (2018) reported $CH_4$ enrichments that were observed during summer in the upper water column of the Gotland Basin, central Baltic Sea. They furthermore found that zooplankton is one but not the only $CH_4$ source in the oxygenated upper waters. While the authors ruled out a major contribution of algae to the observed sub-thermocline $CH_4$ enrichment because of the low sub-thermocline phytoplankton biomass, they considered a primary production associated $CH_4$ formation as one likely source in the phytoplankton-rich mixed layer. The average phytoplankton carbon biomass of the mixed layer was approximately 600 µg $L^{-1}$ (averaged from Fig. 9 in Schmale et al., 2018). For the reported average net $CH_4$ production rate in the mixed layer (95 pmol $CH_4$ $L^{-1}$ $d^{-1}$), we calculated that a production rate of 2.5 µg $g^{-1}$ POC $d^{-1}$ is required if the $CH_4$ is produced by the algal biomass. This rate would be within the range of $CH_4$ production rates observed in our study. These calculations should be considered as a first rough estimate to assess whether $CH_4$ production rates of laboratory grown cultures can significantly contribute to $CH_4$ supersaturation associated with phytoplankton. We did not distinguish between species and did not take into account environmental factors or the complexity of microbiological communities."

*Minor issues*

*Referee #1: Equation 7: There should be a factor of 1000 to convert ratios to per mille values*

Authors: We would like to keep equation 7 as is as it follows the recommendations by Coplen (2011) ("Guidelines and recommended terms for expression of stable isotope-ratio and gas-ratio measurement").

*Referee #1 Figure 1: Plot control values here too.*

Authors: Control groups in Figure 1 were plotted in the revised manuscript. In order to add control groups the unit was changed in concentration (ng $CH_4$ $L^{-1}$).

*Referee #1 Line 268: Should it be "were applied" not "where applied"?*

Authors: Yes. Corrected.

*Referee #1 Line 308: Inoculation OF cells?*

Authors: Yes. "of" was added.

**References**

[revised manuscript text omitted]

**Point-by-point response to the issues raised by referee #2 (Anonymous)**

We thank the reviewer for efforts in reviewing our manuscript and for the helpful comments which have improved the manuscript.

*Referee #2 (referee's comments are in italics)*

*The present paper presents an interesting study about methane production under oxic conditions in marine environments. This so called "methane paradox" is a very important research field to understand methane emissions from oceans (and lakes) and has recently received strong interest by a number of investigators from different scientific disciplines. The author presents*
*data from incubation experiments conducted with three different algal species. Methane production rates were determined with different methanogenic substrates ($^{13}C$-labled) using a stable isotope approach. A similar kind of studies was previously conducted for Emiliania huxleyi by Lenhart et al. (2016) and the isotope approach was successfully used in diverse investigations by Frank Keppler before to examine terrestrial methane production. The novel outcome in the present study is (1) that also other widespread haptophytes have the potential to produce methane under anoxic conditions; and (2) methylated*
*sulphur compounds (e.g. DMS), that are known to be enriched in the investigated algae species, present potential substrates. In addition, the authors present an attempt to transfer their results to an algal bloom in the Pacific Ocean to discuss the potential relevance of algal methane production.*

*The experiments are well thought out and the results present an additional piece in the complex puzzle. There are lots of little corrections needed and from my point of view some sentences need another structure to make the content more accessible for*
*readers that are not familiar with the topic in detail (especially in the method section, e.g. PP, exponential growth rate). I will give a few examples below.*

*Some minor and major points need to be addressed and I therefor recommend a publication after major revision.*

Authors: We thank the referee for the positive evaluation of our manuscript and for the helpful comments. Requested changes were taken into account, as detailed in the following.

*Referee #2: Line 95ff. The experimental design is very complex. A flowchart for the method section would be helpful for the reader.*

Authors: We added a graphic/flowchart to the method section of the revised manuscript.

*Referee #2: Line 98ff. How clean are the algal culture samples (purity)? Small differences in the degrees of contamination with archaea/bacteria (nitrogen limited bacteria, Line 69,Damm) between the cultures may have an impact on $CH_4$ production rate. Does the web link give information about the purity of the culture?*

Authors: Unfortunately, the weblink gives no detailed information about the purity of culture. We cannot consider our approach as axenic because it is extremely difficult to grow algal cultures without any bacteria. However, the algal cultures were diluted regularly, resulting in exponential algal growth and minimal bacterial growth. This is a common practice to keep non-axenic algae cultures largely free of bacteria (please see also answer to comment of reviewer #1 concerning manuscript line 98).

We now mention that this cultures were non- axenic. We added the following sentence: "In order to keep non-axenic algae cultures largely free of bacteria, the cultures were diluted regularly, resulting in quasi constant exponential algal growth while minimizing bacterial cell density."

However, we are aware that bacteria might play a role in $CH_4$ production, but even if they did they still would depend on algal growth in our cultures as demonstrated by the following points.

**1)** The $CH_4$ production rates decreased with decreasing algal growth rates:

In batch cultures, the algae cultures undergo various stages of growth (see section 3.1, Fig. 1 a-c). Bacterial density increases tremendously when algae culture reach stationary growth phase and excretion of organic products from senescent alga cells together with the decomposition of cells is providing substratum for heterotrophs. This was described in literature (Salvesen et al., 2000) and is in line with our own experience with growing alga cultures in batch mode. In section 3.1 cultures have undergone transitionary growth phase leading up to the stationary phase. We calculated daily incremental $CH_4$ production rates (not shown in the manuscript). The $CH_4$ production rates of each species decreased with decreasing growth rates and decreased drastically when approaching stationary phase. This observation is the opposite of what we would have expected, if $CH_4$ were mainly produced by bacteria. It would however be compatible with the idea that algae produce precursors which are subsequently used by bacteria to produce $CH_4$.

**2)** Light is a prerequisite for $CH_4$ formation in algae cultures:

Cultures of *E. huxleyi,* and *P. globose* were incubated under a day-night-cycle and continuous darkness. Methane concentrations did not increase when cultures were incubated in darkness while concentrations increased in cultures growing under a day-night-cycle. This is a strong indication that $CH_4$ formation is dependent on the light-dependent metabolism of the algae, since the metabolism of heterotrophs or archaea is independent of light. While the latter conclusion does not rule out the "algae precursor scenario", our experimental setup makes it rather unlikely. In these experiments we inoculated high cell densities ($\approx 10^5$ cells mL$^{-1}$) because they were designed to be short term which requires a high start cell density to yield measurable production. Therefore the start conditions will have included a seawater replete with precursors. It is unlikely that the relatively few bacteria present should have become precursor-limited over a single dark phase. It is rather more likely that the pool of precursors was sufficient to sustain bacterial $CH_4$ production over the dark phase. In this scenario an extra precursor production by cultures exposed to light would have been without effect on $CH_4$ production.

**3)** It is highly unlikely that methanogenic ***archaea*** are the source of $CH_4$ in cultures where $CH_4$ is produced alongside oxygen (incubation under day-night-cycle).

If archaea were the $CH_4$ source we would have expected a higher $CH_4$ production in the dark.

**4)** Selectively inhibition of algal growth reduced $CH_4$ production rates:

We compared emission rates of *E. huxleyi* that have been treated with and without 3-(3,4 dichlorophenyl)-1,1-dimethyl-urea (DCMU). DCMU acts as an inhibitor of photosynthesis (Wessels and Van Der Veen, 1956). Selectively inhibition of algal photosynthesis reduced both algal growth rate and $CH_4$ production rates. In the inhibition experiments, the growth rate was only 29% of the uninhibited culture and the $CH_4$ production rate dropped to 18% of the uninhibited culture. Since the inhibition effect of DCMU is very selective for algae (Francoeur et al., 2007) the result may indicate direct $CH_4$ production from algae. Although we regard it as unlikely, we cannot strictly rule out the "precursor-scenario": Bacteria use algae-derived precursors to produce $CH_4$, and these bacteria require constant production of these precursors by algae. In other words, the precursor-production by algae is the rate limiting step of $CH_4$ production by bacteria (as evident from points 1, 2, and 4 above). If true the $CH_4$ production observed in our experiments would be the result of a "collaborative effort" which needs both partners, algae and bacteria. This would be a significant finding and prompt further research. Questions to be addressed would include: what are the precursors? Which algae can produce the precursors? Which bacteria can produce $CH_4$ using these precursors? Is it possible to grow the respective algae without the bacteria (not all algal cultures can survive in an axenic state). Can the same $CH_4$ production be achieved by growing the bacteria without algae and adding the precursors? This selection of questions would suffice for an entire research project. Meanwhile we are content with describing $CH_4$ production that depends on algae, whether solely or in cooperation with bacteria.

To sum up, our ***main finding*** is that $CH_4$ production in ***mixed algae/bacteria cultures*** depends on ***algal growth*** and is not supported when algae become senescent. Future research will clarify whether algae alone produce $CH_4$ or whether they produce precursors which in turn are used by bacteria to produce $CH_4$. We have made this important point clear in the revised manuscript.

We added this information in supplementary material. We now discuss the possible contribution of bacteria and archaea in the main text and refer to further discussion in the supplementary material (see discussion above). The revised the paragraph (Chapter 4.1) now reads:" The algal metabolites DMSP, DMS and DMSO are ubiquitous in marine surface layers and nanomolar concentrations were found in blooms of *Chrysochromulina sp., P. globosa* and *E. huxleyi.* Several field studies showed that these compounds are linked to $CH_4$ formation in seawater (Damm et al., 2008; Zindler et al., 2013; Florez-Leiva et al., 2013). The authors proposed that DMSP and their degradation products DMSO and DMS are used by methylotrophic methanogenic archaea, inhabiting anoxic microsites, as substrates for methanogenesis. In addition it was reported that, if nitrogen is limited but phosphorus is replete, marine bacteria might also use DMSP as a carbon source, thereby releasing $CH_4$ (Damm et al., 2010).

One scenario which we cannot rule out would be a production of $CH_4$ precursors by algae and a usage of these precursors by bacteria to produce $CH_4$. While we think that this is less likely than $CH_4$ production by algae alone, it would, if true, show that bacteria need algae-produced precursors to produce $CH_4$. The latter scenario would be relevant in the field because algae co-exist with bacteria in the oceans. Therefore bacteria might be involved in the $CH_4$ production process in our cultures, but even if they were they still would depend on algal growth. For further discussion of a potential contribution of heterotrophs and/or methanogenic archaea see supplementary material. The correlations we describe in the supplementary material clearly show that $CH_4$ production depends on algal growth. It is therefore highly unlikely that bacteria are solely responsible for $CH_4$ production in our cultures."

*Referee #2: Line 133. What is the difference in concentration of $NaHCO_3$ between natural and inoculated water sample? Why did the authors added this amount of tracer? Should be mentioned.*

Authors: Natural North Sea surface seawater contains ca 2000 µmol $L^{-1}$ bicarbonate. We added 48,7 µmol $L^{-1}$ $^{13}$C-bicabonate , i.e. about 2 % of the natural concentration. This bicarbonate concentration was chosen for two reasons, one analytical, the other physiological. The physiological reason is that phytoplankton is sensitive to changes in seawater carbonate chemistry (see reviews on "ocean acidification"). We aimed at a negligible physiological effect of the added bicarbonate. The chosen bicarbonate concentration fulfills this criterion. The analytical reason is this: On the basis of the amount of added $^{13}$C-bicarbonate we calculated the theoretical $\delta^{13}$C -DIC value (see also manuscript line 134). Based on the theoretical $\delta^{13}$C -DIC value and from the previously determined $CH_4$ increases in the cultures, the $\delta^{13}CH_4$ values can be estimated. The amount of $^{13}$C- bicarbonate was chosen on the basis of expected changes of $\delta^{13}CH_4$ values which were measured using GC-IRMS. A change of tenth to few hundred per mil is ideal regarding statistical issues (applying keeling plots for source identification) but also concerning linearity issues of the isotope ratio mass spectrometer.

*Referee #2: Line 141. Here, you should explain in more detail why an exponential growth rate is important to best compare $CH_4$ formation between the experiments. From this sentence one could assume that Langer performed already methane production rate experiments with algae that indicated that exponential growth rates are important. From my point of view the activity of the cell is important for the turnover of these substrates and not their reproduction. It should be mentioned in the method or result section that microbial methane turnover takes place in the incubations and the production rates presented are minimum rates > because methane oxidation is not considered in the calculations (e.g. see methods in de Angelis and Lee).*

Authors: We agree with the reviewer that from a physiological point of view the activity of the cell is the relevant parameter here. But as detailed below our point is purely methodological, not physiological.

We have clarified this in the revised manuscript:

"Exponential growth is a prerequisite for calculating production on the basis of growth rate and quota (here $CH_4$ quota). The point is a general, technical one, and is not confined to $CH_4$ production. The studies by Langer et al. (2012, 2013) discuss this point in the context of batch culture experiments. Briefly, production on this account is the product of growth rate and quota (e.g. $CH_4$, calcite, organic carbon). Production here is an integrated value, typically over many cell divisions. For this calculation of production to be meaningful a constant growth rate is required. The exponential growth phase fulfills this criterion whereas the transition phase and the stationary phase do not. Therefore production cannot be calculated meaningfully in the non-exponential phases. The problem can, however, be minimized by using small increments (one day) because growth rate can be regarded as quasi-constant (see also Lenhart et al., 2016). "

We agree with the reviewer that our calculated rates should be regarded as minimum rates because of *microbial methane turnover*. Please see also answer to comment on manuscript line 98ff.

*Referee #2: Line 137. Can you explain if aggregates or sediment was visible in the incubation?*

Authors: Cells were counted under bright field microscopy and we did not observe any aggregates.

*Referee #2: Line 172. Why was exactly this amount of substrate (DMS...) injected and is this comparable with natural environments (concentrations). Substrate concentrations definitely affect the turnover and the addition of tracers/substrates should not impact the sample too drastically. Why did the authors did not applied MET (and DMSP) as a precursor that was*

*tested before successfully by Lenhart et al.?*

Authors: The amount added was chosen based on the practical experience from previous experiments with *E. huxleyi* and methionine (Lenhart et al., 2016), so that the expected $\delta^{13}CH_4$ fall within a measurable range with statistical significance. The growth of algae was not effected and changes of the overall $CH_4$ production did not change by the addition of substrate (within error of measurement).

The added amount (10µM) of methylated sulfur compounds ($^{13}C_2$-DMS, $^{13}C_2$-DMSO or $^{13}C$-MSO) was higher than those expected in ocean water samples (please see also answer to reviewer #1 comment on manuscript line 133). However, the intracellular concentrations of these compounds can reach mM levels (Keller, 1989; Rafel et al.,1998; Keller et al., 1999; Sunda et al., 2002), which is two orders of magnitude higher than the added concentration of 10 µM (final) in our experiments. For this reason, it can be expected that the amount of $^{13}C$ labelled substance taken up by the algal cells is low in relation to the amount of methylated sulfur compounds what they synthesize during metabolism.

The turnover of DMS, DMSO or MSO (including non-labeled compounds) to $CH_4$ could not be determined on the basis of their added amount of $^{13}C$-labelled substance in cultures of *E. huxleyi*. Please see answers regarding comments of reviewer#1 (manuscript 327), where we discuss this issue in detail.

Neither was this the goal of the experiment. However, it can be determined exactly how the ratio of $^{13}C$ in $CH_4$ increases, when

$^{13}C$ labeled methylated sulfur compounds were added (Fig. 4 a-c). It has therefore been shown that methyl groups of these compounds can be converted into $CH_4$ in algal cultures.

Due to time constraints, we omitted the methionine treatment. Unfortunately, isotopically labelled $^{13}C_2$ DMSP was not commercially available. Moreover, this compound could not be synthesized in our laboratories.

*Referee #2: Line 327ff. Is it possible that a natural microbial community is needed for the turnover of these substances to methane? If the incubations are without contaminations (sterile filtered seawater, pure culture), the production rates might be low because of the missing community. The algae may only provide the precursors. Might be a point that could be discussed here.*

Authors: Please see reply to referee #2: line 98ff .

**Referee #2: Line 381.** *Argumentation is difficult. Only because Lenhart could prove a contamination-free incubations, this result cannot be transferred to all the incubation that will be performed by the working group afterwards. See comments/concerns to this topic above. Since the argumentation is difficult to follow, I suggest to discuss this topic less dominant and integrate this part somewhere else (not under a separate title). Also 50% of the text is nearly copied from the*

*introduction (doubling!).*

Authors: Agreed. This section has been modified according the referee's suggestions. Please see also reply to referee #2: line 98ff.

**Referee #2: Line 335.** *I have a different impression. Figure 4a: At day 2 the d13C values are very close to each other. In*

*Figure 4b all values from beginning to the end of the incubation time are very similar. Only Figure 4c shows a clear difference between culture and control over the course of the experiment. Add in the figure caption that also controls are plotted, not only results from cultures.*

Authors: $\delta^{13}CH_4$ values for $^{13}C_2$-DMS are presented in Figure 4 a. We found that DMS is also converted chemically in sterile filtered seawater. This is in line with observations of Zhang et al. (2015) (please see manuscript line 332-339). However, the formation rates are very low and only become obvious when applying sensitive stable isotope labelling techniques. We agree with the reviewer that the $\delta^{13}CH_4$ values of the DMS spiked seawater group and the DMS spiked algae group are very close to each other up to day 2. A section regarding control values was added to the revised manuscript (Chapter 4.1) :" The $^{13}C_2$-DMS spiked seawater group and the $^{13}C_2$-DMS spiked algae group are very close to each other up to day 2 (see Fig.3a and Fig.4a). For this time period, it can be assumed that the chemical conversion has taken place in both samples to the same extent, since the samples are relatively similar, because the algal cell density is only 5% (day 2) of the final cell density. However, the following days (day 3 to day 6), when algal cell density increased drastically, the $\delta^{13}CH_4$ values of the algae cultures also increased significantly compared with $\delta^{13}CH_4$ values of the seawater. This clearly indicates that conversion of $^{13}C_2$-DMS to $CH_4$ increases with increasing cell counts."

**Referee #2: Chapter 4.3** *I would recommend to perform an additional calculation to show that algal CH₄ production is an*

*important mechanism that can explain air/sea methane fluxes and methane enrichments. For example Schmale et al. (2018) gives detailed data about phytoplankton biomass (e.g. Prymnesiales) and production rates needed to maintain air/sea fluxes and subthermocline methane enrichments. There are probably also other papers available that could be used for such calculation.*

Authors: We followed the recommendation and used the detailed data of Schmale et al. (2018) to estimate a possible contribution of algal $CH_4$ production to the $CH_4$ production rate in the field. An additional calculation was added to Chapter 4.3 in the revised manuscript. See reply to referee #1: line 400.

*Minor issues:*

**Referee #2: Title:** *I recommend writing "potential relevance for the environment". A direct transfer*
*of laboratory studies/results into field observations is difficult.*
Authors: Change applied.

**Referee #2: Line 24.** *Please also give the productions rates per cell in the abstract. Temperature is*
*not needed to be mentioned in the abstract.*
Authors: Changes applied.

**Referee #2: Line 27ff.** *It should be mentioned here that the conversion of methylated sulphur compounds to methane was only*
*responsible for less than 1% of the observed methane production (line 327ff).*
Authors: We do not think that this information is important for the reader. The information might misleading here. Please see
answers regarding comments of reviewer#1 (manuscript line 327), where we discuss this issue in detail.

**Referee #2: Line 26-29.** *The word "clearly" is used to often.*
Authors: We have rephrased line 26-29.

**Referee #2: Line 30.** *"Relevance for the environment" is one major issue in the title but is reduced*
*here to a little sentence. This part should be extended.*
Authors: We have emphasized this issue in the abstract as requested. We added the following sentence: "By comparing the
algal $CH_4$ production rates from our laboratory experiments with results previously reported in two field studies of the Pacific
Ocean and the Baltic Sea we might conclude that algae mediated $CH_4$ release is contributing to $CH_4$ oversaturation in oxic
waters. Therefore, we propose that haptophyte mediated $CH_4$ production could be a common and important process in marine
surface waters."

**Referee #2: Line 49**. *How can "emissions from freshwater" explain the CH4 concentration in ocean*
*surface water?*
Authors: We deleted "emissions from freshwater and" from line 49.

**Referee #2: Line 50.** *Shorten the sentence and delete "that has been often: : :".*
*"Well-known" means "often reported"*
Authors: We have appropriately modified the sentence.

*Referee #2: Line 55-58. This paragraph should be moved to line 46. It might be better to start with this overall review before listing the recent specific studies to oxic methane production*

*in lakes and ocean.*

Authors: As requested, we have restructured this section.

*Referee #2: Line 60. May also mention Valle and Karl (2014) who used in situ MPn concentration in a 14C approach and showed that dissolved MPn in surface waters cannot account for methane oversaturation.*

Authors: The revised manuscript contains the results of del Valle and Karl (2014) as well as the results of Repeta et al. (2016), who, on the other hand, reported that the cycling of the organic matter phosphonate inventory could be sufficient to support the total atmospheric $CH_4$ flux at their study site.

*Referee #2: Line 98. A bracket is missing (RC: : :). Is it clear for the reader for what the web link is*

*good for?*

Authors: The bracket was added. We have added the names of the culture collections to the front of the weblink to make it clear that the links lead to the respective collections.

*Referee #2: Line 102. Delete "in" in front of "natural"*

Authors: "in was" deleted.

*Referee #2: Line 110ff. Why did the authors used different volumes (medium and headspace)?*

Authors: For practical reasons. We have limited space in the climatic chamber and a limited supply of natural seawater, therefore the size of the vessels were adjusted.

*Referee #2: Line 119. What is meant with "main cultures"? Is this the investigated culture in the*

*incubation?*

Authors: Yes, it means the culture that was studied during the incubation. We rephrased this sentence to make it clear.

*Referee #2: Line 119ff. I would suggest to transfer the cell densities to the result section (3.1).*

Authors: Changes applied as suggested.

*Referee #2: Line 122. E. huxleyi was sampled daily! What do you mean with overall sampling*

*interval: 9,11,6 days? Why this odd order? And why did you sample the cultures in different intervals? If there is a reason for that it should be explained.*

Authors: Overall sampling interval means the incubation time (from inoculation to the end of the experiment/incubation) for each species, that correspond to the sampling time. The incubation time varies from species to species and depends on the growth rate and the cell density in the stationary phase. The stationary phase for each species is dependent on a species specific cell density. As the species grow at different rates the sampling intervals differ. *E. huxleyi* has by far the highest growth rate and was sampled daily. *Chrysochromulina sp.* and *P. globosa* grow slower and were sampled at longer intervals. We revised this section in the manuscript and provided explanations.

*Referee #2: Line 128. Why only three data points for E. huxleyi. From Figure 1c it seems to be plausible to use four.*
Authors: The phase of exponential growth (from which μ was calculated) was defined by the cell densities which correspond to the best fit of the exponential regression. The fourth data point clearly deviates from exponential regression.

*Referee #2: Line 131. It is always worth to have a repetition to support the previous results.*
Authors: Due to the time constraints, we decided not repeat the experiments with stable isotope measurements that were already done by Lenhart et al. (2016).

*Referee #2: Line 134. The delta is missing in d13C*
Authors: Corrected.

*Referee #2: Line 135. Suggest to write: ": : :values of the methane precursor: : :"*
Authors: The expression "source" is usually used in combination with Keeling plots. We would therefore like to keep the expression "source".

*Referee #2: Line 144. ": : :measured at the end: : :"*
Authors: Corrected.

*Referee #2: Line 145. Suggest to write "For this additional experiment: : :"*
Authors: Agreed. Changes applied.

*Referee #2: Line 146. Suggest putting the cell densities in the result section (see above).*
Authors: Changes applied as suggested.

*Referee #2: Line 153. "ag" is the abbreviation for what?*
Authors: Please see bracket in line 153. The unit "ag" means $10^{-18}$ g. The SI prefix "a" stand for atto ($10^{-18}$).

*Referee #2: Line 158. The program "Image J" is produced by which company?*

Authors: "Image J" is an open source software. A reference was added.

*Referee #2: Line 174. I still think that cell densities should be implemented in the result section (see above).*

Authors: Changes applied as suggested.

*Referee #2: Line 176. The target/design of the experiment in section 3.2 is still unknown!*

Authors: We have revised section 3.2 to make the experimental target and design clearer. Please see also the answer to the comment concerning manuscript line 95ff.

*Referee #2: Line 189. Analyzed. See line 180*

Authors: Corrected.

*Referee #2: Line 193. Write: ": : (based on three: : :)".*

Authors: We changed line 193 as requested.

*Referee #2: Line 204. Delete: ": : :at a temperature: : :". Here and in some other parts of the result section you mention*

*details that were mentioned before in the method section. I would start with a sentence that makes clear that you are talking about the incubation with $^{13}$C-labelled hydrogen carbon (2.3).*

Authors: Line 204 "at a temperature" was deleted. The result section was revised to avoid repetition of details from the method section. An introductory sentence was added in section 2.3 as suggested.

*Referee #2: Line 208. Also here delete the repeated information: "These rates were obtained: : :".*

*Check the entire result section to avoid redundancy.*

Authors: The result section was revised to avoid repetition of details from the method section.

*Referee #2: Line 211. The cell density should only be mentioned here and not in the method*

*section!*

Authors: All final cell densities were removed from the method section.

*Referee #2: Line 214. Where is the control group plotted? Figure 2. Black and blue dots are difficult to distinguish. Even if it is "only" the control sample – make the visibility easier. The x-axis should be 1/CH4. Right?*

Authors: Control groups in Figure 1 were plotted in the revised manuscript. Please also look at the answer to referee#1's comment regarding figure 1. Figure 2 was revised: by changing colors and correcting labeling of x-axis. Yes, "1/CH$_4$" is correct.

    ***Referee #2: Line 249***. *See above. Not clear why the exponential growth phase is important and not the cell activity.*

Authors: Please see the answer above to the comment concerning manuscript line 141.

    ***Referee #2: Line 251.*** *The equation is already described above (2 and 4). Avoid doubling. See comment above.*

    Authors: We revised this section in order to avoid doubling. The sentence "By doing so the CH$_4$ production rate is the product of exponential growth rate μ and cellular or POC quota." was removed from line 251.

    ***Referee #2: Line 254.*** *The sentence should end with (Tab. 1).*

    Authors: Corrected.

***Referee #2: Line 256.*** *"community level" sounds odd in this context. May you can find a better description.*

    Authors: This term was used in the context of the production potential which was established by Gafar et al. (2018). The phrase "community level" is also used in this context by Gafa et al. (2018). We would therefore like to keep this phrase in order to avoid misunderstandings.

    ***Referee #2: Line 266.*** *It starts again with information that was mentioned before in the methods.*

    Authors: We removed this information from line 266.

    ***Referee #2: Line 271***. *The sentence should end with "(Fig. 4)".Figure 4b. Change the x-axis to 13C2 (add 2).Figure 4c.*

*Change the x-axis to MSO (not MES, see caption).*

    Authors: Line 271 and Figure 4c were corrected.

    ***Referee #2: Line 279***. *Add the control sample in the Figure.*

    Authors: Control groups were added.

    ***Referee #2: Line 302***. *In the present study only the turnover of [13]C-hydrogen carbonate by two algal species was investigate. Lenhart applied the isotope technique for E. huxleyi.*

    Authors: We have corrected this sentence.

***Referee #2: Line 307.*** *(with highest cell numbers) is out of context. Please rephrase.*

    Authors: The phrase "(with highest cell numbers)" was replaced by "(where the POC content is highest)".

    ***Referee #2: Line 333***. *In future investigations I would suggest a dark incubation to exclude methane*

    *production by UV or visible light (line 70ff).*

Authors: This topic is currently being investigated by us. Please see also answer to the comment of referee #2 concerning

    manuscript line 98. There we discuss a dark incubation experiment. In addition we discuss there a experiment with and without

    inhibition of algae by DCMU.

    ***Referee #2: Line 352ff.*** *Did Althoff really proved that the "reactivity" is the driving force in her experiments? Or are point 1*

*(label concentration) and 2 (penetration) also possible explanations*

    *for her observations?*

    Authors: Althoff et al. (2014) used a defined chemical system to study the conversion of methylated sulfur compounds to $CH_4$.

    It turned out that the yield was not the same for all substances under otherwise identical reaction conditions. The $CH_4$ yield

    was therefore also dependent on the substance and thus its reaction behavior.

    ***Referee #2: Line 360ff.*** *Sentence too complex. Devide in two parts.*

    Authors: As requested, we have reformulated and restructured this sentence.

    ***Referee #2: Line 363*** *and 365. Too often "furthermore". Rephrase.*

Authors: Changes applied.

    ***Referee #2: Line 391***. *Interesting. But it needs be explained in more detail why the growth rate*

    *impacts the methane production. See above.*

    Authors: A detailed explanation was added to the method section. Please see reply referee #2: line 141.

    ***Referee #2: Line 411.*** *Include/explain why PP is meaningful parameter.*

    Authors: Further explanation regarding the meaning of PP was added to the manuscript. Line 410 now reads: " However,

    several recent studies have emphasized that the production potential (PP), as opposed to cellular production, is a

    biogeochemically meaningful parameter (Gafar et al., 2018, Marra 2002, Schlüter et al., 2014, Kottmeier et al., 2016). The concept of the production potential goes back at least to the first half of the 20[th] century (Clarke et al., 1946). Briefly, the

    production potential of substance X is the amount of X which a phytoplankton community or culture produces in a given time.

For details see Material and Methods and references above. The cellular production by contrast is the rate of production of X of a single cell, and therefore the cellular production is ill qualified to express community-level production."

For detailed information about changes made to the manuscript please see point by point response to the referee and track changes version of the manuscript.

[revised manuscript text omitted]

**Experimental set up**

determination of
CH₄ concentration
and cell density determination of stabel carbon
isotope values of CH₄

investigation groups (n=6):

investigation groups (n=3):

culture medium + culture + H¹³CO₃⁻

medium + culture medium medium + H¹³CO₃⁻

medium

**Fig. 2 Experimental setup for measuring CH₄ formation by *Chrysochromulina sp.* and *P. globosa*. Methane formation was investigated by concentration measurements within six vials containing either algae or medium only (left column). For stable isotope measurements of CH₄ ¹³C labelled hydrogen carbonate (H¹³CO₃⁻) was added to three vials of both groups (right column).**

The overall incubation time –was 9, 11 and 6 days for *Chrysochromulina sp., P. globosa* and *E. huxleyi* respectively. Hheadspace and liquid samples were collected on a daily basis for *E. huxleyi* and in 2-3 days intervals from cultures of *Chrysochromulina sp.* and *P. globosa*. The incubation time and sampling intervals varied between species because of variations in the growth rate and the cell density in the stationary phase.

Cell densities were plotted versus time and the exponential growth rate ($\mu$) was calculated from exponential regression using the natural logarithm (Langer et al., 2013). The  exponential growth phase (from which $\mu$ was calculated) was defined by the cell densities which corresponded to the best fit ($r^2 > 0.99$) of the exponential regression. This was done by using the first three (*Chrysochromulina sp.* and *E. huxleyi*) or four data points (*P. globosa*) of the growth curve.

For stable carbon isotope experiments 48,7 µmol L$^{-1}$ $^{13}$C-hydrogen carbonate (NaH$^{13}$CO$_3$) in final concentration was added to the F/2 medium. The added amount of NaH$^{13}$CO$_3$ corresponds to 2% of the DIC of the North Sea seawater (2152 ± 6 µmol L$^{-1}$), resulting in a theoretically calculated δ$^{13}$C value of DIC of +2014 ± 331‰. To determine the δ$^{13}$C-CH$_4$ values of the source, the Keeling-plot method was applied (Keeling, 1958). For a detailed discussion of the Keeling plot method for determination of the isotope ratio of CH$_4$ in environmental applications, please refer to (Keppler et al., 2016). Oxygen concentration was monitored daily (using inline oxygen sensor probes, PreSens, Regensburg) at the end of the light cycle (Fig. S1).

**2.4 Determination of CH$_4$ production rates**

Since the experiment in the section 3.2 2.3 was not designed to obtain POC quotas (POC = particulate organic carbon), we conducted an additional experiment. To best compare CH$_4$ formation rates of the three algae species it is necessary to obtain exponential growth to ensure constant growth rates and constant (at a given time of day) cellular POC quotas over the course of the experiment. (Langer et al., 2012, 2013). Exponential growth is a prerequisite for calculating production on the basis of growth rate and quota (here CH$_4$ quota). The point is a general, technical one, and is not confined to CH$_4$ production. The studies papers by Langer et al. (2012, 2013) discuss this point in the context of batch culture experiments. Briefly, production on this account is the product of growth rate and quota (e.g. CH$_4$, calcite, organic carbon). Production here is an integrated value, typically over many cell divisions. For this calculation of production to be meaningful a constant growth rate is required.

The exponential growth phase fulfills this criterion whereas the transition phase and the stationary phase do not. Therefore production cannot be calculated meaningfully in the non-exponential phases. The problem can, however, be minimized by using small increments (one day) because growth rate can be regarded as quasi-constant (see also Lenhart et al., 2016). The

CH$_4$ production rates can be calculated by multiplying the growth rate µ with the corresponding cellular or POC-CH$_4$ quota, that was measured on at the end of the experiment.

For this additional experiment t The cultures were grown in 160 ml mL crimped serum bottles filled with 140 ml mL medium and 20 ml mL headspace (n=4). The initial cell density of 22.5 ± 3.1 × 10$^3$ cells mL$^{-1}$, 80.9 ± 11.5 × 10$^3$ cells mL$^{-1}$ and 29.0 ±

[revised manuscript text omitted]

---

## Author Response (AR2)

Dear Prof. Dr. Jack Middelburg

On behalf of all authors, I would like to thank you for the excellent editorial handling and for reviewing our manuscript.

With best regards,

Thomas Klintzsch

**Point-by-point response to the issues raised by the editor (Jack Middelburg)**

*Editor: (editor comments are in italics) While reading, I identified the following corrections/potential improvements:*
*- Write out species in full upon first use, e.g. T. pseudonana in l.89.*

Authors: In the revised manuscript species names were written out in full upon first use in the abstract and then again at the
10  first use in the rest of the text. The abbreviated spelling is now continuously used.

*Editor: - Line 42: challenges*
*- L. 44: replace On the other hand with However because there is no on the one hand*
*- L. 50, cyanobacteria, as CH4 producers, suggesting*
*- L. 58: add space*

15  Authors: Changes applied.

*Editor: L. 64: cycling of the organic matter phosphonate inventory: please reformulate, unclear because too cryptic*

Authors: We removed this information from the manuscript. The sentence (line 62) was modified to now read: " In addition, it has been suggested that $CH_4$ might be produced under phosphorus limitation by bacterial cleavage of methylphosphonate (MPn) in oligotrophic Pacific regions (Karl et al., 2008; Metcalf et al., 2012; Repeta et al., 2016; Valle and Karl, 2014)."

20  *Editor: L. 81-82: rewrite more actively.*

Authors: The two sentences were modified to now read: "Under highly oxidative conditions nonheme iron-oxo (IV) species catalyse $CH_4$ formation from methyl thioethers and their sulphoxides (Althoff et al., 2014; Benzing et al., 2017). Iron-oxo species are intermediates in a number of biological enzymatic systems (Hohenberger et al., 2012)."

*Editor: Sections 2.3 and 2.4 in particular, but all through, please do not use too many one or two line paragraphs.*

25  Authors: several paragraphs were combined in the revised manuscript.

*Editor: - Fig. 1: type: stable not stabel.*
*- L. 154: use . rather than , for digits*
*- L. 198: was used for each compound*
*- L. 229: remove brackets from references*
30  *- Fig. 3, use . rather than , for digits (r2)*
*- L. 283: were grown up*
*- L. 403-404: homo oxo iron (IV) has been identified … intermediate…*
*- L. 424: even if they were, they still…*

Authors: All points were corrected.

[revised manuscript text omitted]